# FinThink: An LLM-Based Multi-Agent System for Financial Reasoning

## Abstract

The efficacy of Multi-Agent Systems (MASs) in finance is fundamentally constrained by their reliance on static workflows and simplistic learning paradigms, which fail to address the complex, often contradictory signals inherent in dynamic markets—a reality well-described by the Adaptive Markets Hypothesis (AMH). To bridge the gap between prevailing agent architectures and this market reality, we introduce FinThink, an novel MAS framework grounded in AMH, designed specifically to master this dynamic adaptation. FinThink's novelty lies in three synergistic components: (1) A **Context-aware Workflow for Reasoning (CWRM)**, which enables architectural adaptivity by dynamically adjusting reasoning depth based on signal complexity; (2) A **Reasoning-Driven Hierarchical Memory (R-Mem)**, which facilitates evolutionary adaptivity by allowing the system to learn optimal strategies for resolving signal conflicts under varying market conditions; and (3) A **Sentiment-To-Logic (STL) Prompt Protocol**, which ensures reasoning stability by preventing the multi-agent process from degenerating into simplistic voting. In extensive backtests on five major technology stocks (AAPL, GOOG, MSFT, TSLA, and AMZN), FinThink demonstrates significant improvements over contemporary LLM-based agents, achieving a median advantage of a **9.2%** higher Sharpe Ratio, a **113.1%** higher Calmar Ratio, and a **46.3%** reduction in Maximum Drawdown.

## 1 Introduction

Large Language Models (LLMs) have demonstrated remarkable capabilities in reasoning (Brown & et al., 2020; Wang et al., 2023; Wei et al., 2022; Yao et al., 2023b), planning (Hao et al., 2023; Zhao et al., 2023; Wang et al., 2024; Kokel et al., 2025), and tool use (Gao et al., 2024; Yang et al., 2023; Wen et al., 2025; Gao et al., 2023) across diverse domains (Sanh et al., 2022; Wang et al., 2022; Raffel et al., 2020). This has sparked interest in deploying LLM-based agents for complex real-world tasks that require multi-step reasoning and decision-making under uncertainty (Schick et al., 2023; Zhou et al., 2024; Deng et al., 2023). Multi-agent systems (MASs) have emerged as a natural extension, where specialized agents collaborate to decompose complex problems, cross-validate findings, and leverage diverse perspectives.

In finance, this paradigm is particularly appealing. However, its application faces a fundamental challenge well-captured by the **Adaptive Markets Hypothesis (AMH)** (Lo, 2004), which posits that markets are not static arenas of rational actors but evolving ecosystems of competing strategies. This inherent market dynamism—where signals are frequently conflicting and core profitability factors shift over time—places extreme demands on the adaptability of any intelligent system. Financial decision-making thus inherently involves synthesizing heterogeneous information streams (TETLOCK, 2007; DA et al., 2011), managing uncertainty (Baker et al., 2016; Pástor & Veronesi, 2010), and adapting to these rapidly changing market conditions (Lo, 2004). For a trading agent, at each daily time step $t$, this means observing a set of market information $\mathcal{I}_t$ and producing a trading action $a_t \in \{\text{BUY, SELL, HOLD}\}$ with a position size $q_t$.

Early work has explored domain-tuned foundation models (Lee et al., 2019; cha; Gu et al., 2021; Feng & Guo, 2020) and multi-agent frameworks that emulate human analyst teams (Xiao et al., 2025). However, these systems often exhibit brittle performance when deployed in real market environments. Their failure stems from an architectural mismatch with the market reality described by

**AMH**: they are ill-equipped to handle the adaptive and evolving nature of financial markets (Bailey et al., 2013; Goyal et al., 2024; Khandani & Lo, 2011). Their static reasoning pipelines and reactive, noisy memory mechanisms fail to capture shifting market regimes and distill durable knowledge from transient market signals.

To bridge this gap between existing agent architectures and market reality, we introduce FinThink, an AMH-grounded MAS designed for dynamic adaptation. We instantiate this challenge within the well-defined problem of **tactical single-asset stock trading** (Almgren & Chriss, 2000; Deng et al., 2017), where the goal is to maximize risk-adjusted returns (Ding et al., 2015; Sharpe, 1994). By operationalizing the principles of AMH (Lo, 2004; Kim et al., 2011; Urquhart & McGroarty, 2016; Neely et al., 2009), we can build a robust multi-agent system along two dimensions: architectural adaptivity, where the reasoning workflow reconfigures in response to market context, and evolutionary adaptivity, where the agent distills durable knowledge from experience. This perspective reveals two critical gaps in existing systems that FinThink aims to address:

- **Gap 1: Static Control Under Dynamic Regimes.** Fixed pipelines like "search → reason → act" or Finite State Machines (FSM) with hard-coded transitions lack the ability to modulate reasoning depth when evidence conflicts or to branch to meta-cognition when uncertainty spikes. This creates a need for *context-aware depth control*, in which the agent dynamically adjusts its reasoning depth in proportion to the magnitude of the uncertainty signal (Schuster et al., 2022).

- **Gap 2: Reactive and Noisy Memory.** Trigger-based memory systems, for example FinCon, record events and perform reflection when a drop in Profit and Loss (PnL) occurs (Yu et al., 2024). However, if such a drop merely results from a temporary technical rebound following an overbought/oversold condition, while the overall trend remains intact, this mechanism risks miscoding short-term market noise as structural experience. In contrast, an ideal memory should span the complete life cycle and demonstrate forward-looking utility (Li et al., 2024). Therefore, a critical gap exists for a mechanism capable of distilling generalizable heuristics from entire trading cycles. This requires a process that systematically filters transient market noise to build a robust knowledge base that reliably informs future reasoning.

To tackle the above challenges, we introduce **FinThink**, a MAS designed to fill these gaps by explicitly operationalizing the principles of the AMH. FinThink's novelty lies in the holistic integration of three components, each designed to address a core requirement for success in adaptive markets:

- *To achieve architectural adaptivity*, we introduce a **Context-aware Workflow for Reasoning (CWRM)**, which decouples dynamic planning from static validation: an LLM-based planner *proposes* the next reasoning step and depth based on real-time market signals, while a deterministic FSM *validates* this proposal against formal rules. This symbiotic architecture allows FinThink to flexibly modulate its reasoning depth, deepening analysis during uncertainty and streamlining it during clear trends, thus directly addressing the AMH's call for adaptive behavior.

- *To enable evolutionary adaptivity*, where the system learns from experience, we design a **Reasoning-Driven Hierarchical Memory (R-Mem)**. Instead of reacting to simple triggers, R-Mem performs post hoc analysis on entire trade lifecycles to distill structured *corrective heuristics*. Crucially, we introduce a methodology for organizing these heuristics into **Narrative Driver Clusters** based on empirically-derived asset similarities. This allows FinThink to generalize lessons learned from one asset to its peers, mimicking the expert behavior of evolving a trading "style" rather than just memorizing isolated events.

- *To maintain reasoning stability* in the conflicting and noisy signals characteristic of adaptive markets, we develop a **Sentiment-To-Logic (STL) Prompt Protocol**. The protocol mitigates "Reasoning Collapse" by strictly decoupling cognitive functions: It constrains analytical agents to produce evidence-grounded, logical assessments while prohibiting premature trade recommendations. This ensures that final decisions are the output of a deep, structured reasoning process, rather than a superficial vote, enhancing the system's robustness under pressure.

We evaluate FinThink in challenging open-world tasks, including tactical single-asset trading and cross-asset memory transfer. It consistently achieves superior risk-adjusted returns, driven by its core innovations: a novel context-aware workflow that dynamically generates reasoning paths, and a hierarchical memory that enables evolutionary adaptation. To our knowledge, this work is the

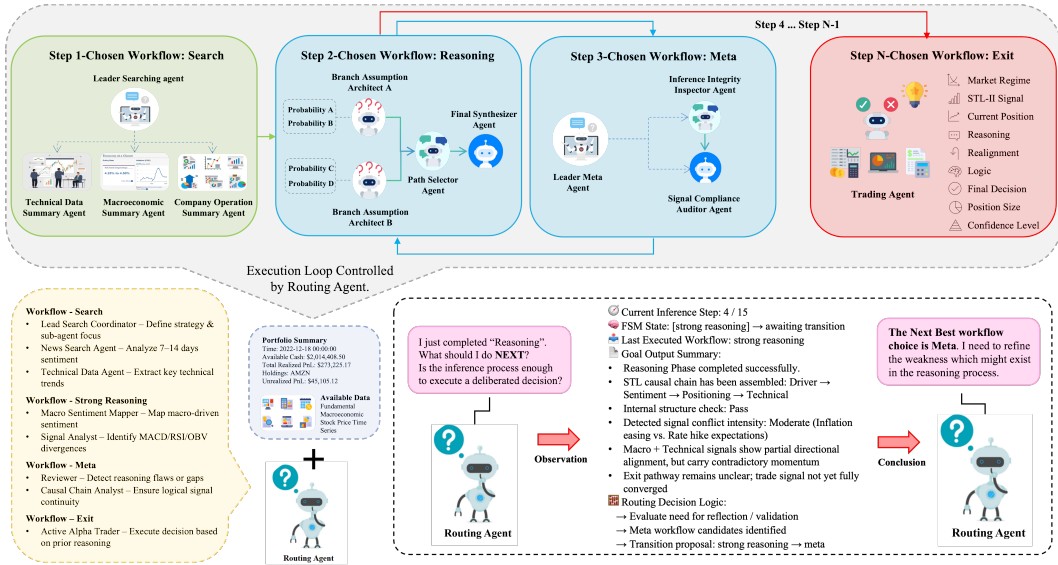

Figure 1: Overview of FinThink's Framework.

first to explore training based on asset-style clustering, which enables a novel form of memory generalization absent in prior work.

## 2 RELATED WORK

### 2.1 MULTI-AGENT SYSTEMS

Recent advancements in LLMs (OpenAI, 2023; OpenAI & et al., 2024; DeepSeek-AI & et al., 2025; Team & et al., 2024; Yao et al., 2023a; Asai et al., 2024) have catalyzed the development of multi-agent systems (MASs). Early approaches focused on collective calibration, using techniques like multi-agent debate (Kim et al., 2024; Hu et al., 2025) and consensus aggregation (Kim et al., 2024) to improve the reliability of single-agent reasoning through redundancy. However, these methods primarily address error correction rather than enabling adaptive collaboration for complex, open-ended problems. To enhance autonomy, subsequent research has explored workflow management. For instance, StateFlow (Wu et al., 2024b) and MetaAgent (Zhang et al., 2025) introduced structured mechanisms like Finite State Machines (FSMs) to orchestrate agent behavior. Similarly, MetaGPT (Hong et al., 2024) and AutoGen (Wu et al., 2024a) developed architectures for automated task decomposition. While advancing structured collaboration, these systems often rely on static, predefined workflows, limiting their adaptability in dynamic environments. Our proposed multi-agent system addresses this gap by introducing a dynamic, context-aware planning layer.

### 2.2 LLM-BASED FINANCIAL AGENTS

In finance, initial research focused on domain-specific models like BloombergGPT (Wu et al., 2023) or fine-tuning open-source LLMs (Liu et al., 2023). To handle complexity, single-agent systems evolved into multi-module architectures like FinAgent (Zhang et al., 2024). Building on this, multi-agent paradigms emerged, with systems like FinCon (Yu et al., 2024) simulating trading teams and HedgeAgents (Li et al., 2025) developing operational risk controls.

Despite these advances, existing financial MAS often suffer from two critical limitations: (1) their static reasoning pipelines struggle to adapt to rapidly changing market conditions, leading to cognitive overload and suboptimal performance (Shang et al., 2025); and (2) risk management is treated as a separate, operational module rather than being integrated into the core strategic reasoning process. In contrast, FinThink directly tackles these challenges by internalizing risk awareness within an adaptive reasoning framework.

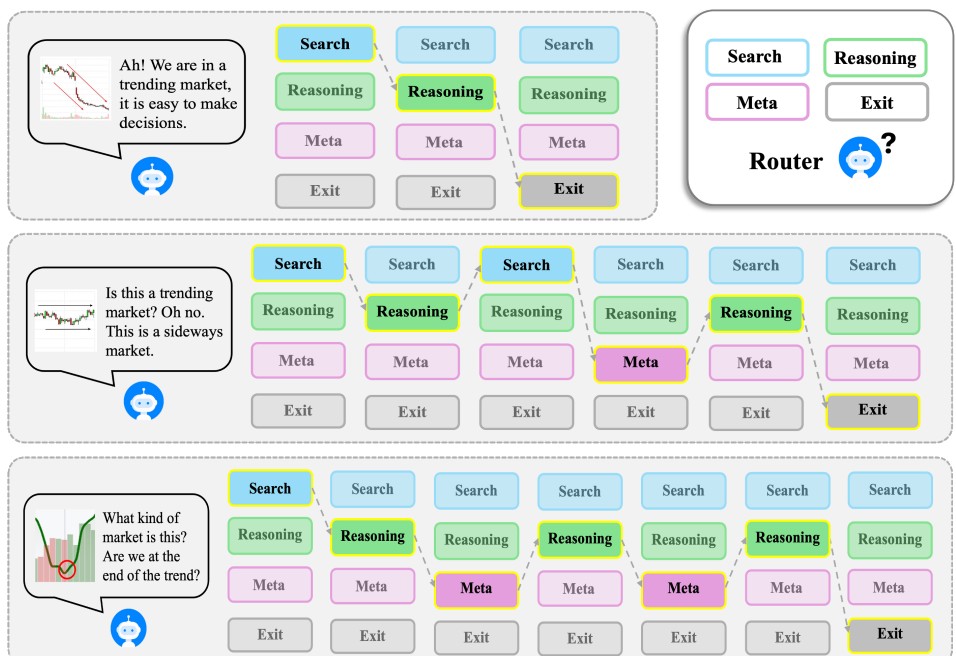

Figure 2: CWRM Routing Process.

# 3 METHODOLOGY

In this section, we present the details of FinThink, which involves the following modules, including Context-Aware Workflow for Reasoning (CWRM), reasoning-driven memory (R-Mem), and Sentiment-To-Logic (STL) prompt protocol.

## 3.1 CONTEXT-AWARE WORKFLOW FOR REASONING (CWRM)

The Context-Aware Workflow for Reasoning (CWRM) framework serves as FinThink's central nervous system (i.e., illustrated in Figure 1), responsible for the dynamic planning, task decomposition, and iterative execution of complex financial decision-making tasks. It deconstructs the intricate financial decision process into a dynamically orchestratable, iterative loop comprising four core workflows: Search Workflow, Reasoning Workflow, Meta Workflow, and Exit Workflow (decision-making). In contrast to traditional frameworks that employ fixed procedures, the essence of CWRM lies in its context-aware dynamic scheduling capability. It intelligently determines which type of workflow to execute next step based on the context of past executions, current market information, ensuring that each action addresses the most pressing cognitive requirement at any given moment.

**Planning - RouterBackbone**: At the commencement of each reasoning node, the Routing Agent, functioning as the system's "decision-making brain", is initially activated. It comprehensively assesses the current context, encompassing: (i) Historical Context: Outputs and conclusions from preceding workflows (e.g., Search Workflow, Reasoning Workflow). (ii) Market Context: The latest market snapshot and investment portfolio status. (iii) Current State: The current state as recorded by the Finite State Machine (FSM). Based on this comprehensive evaluation, the Routing Agent plans and proposes the most appropriate workflow to execute the next step.

This dynamic scheduling capability is visually demonstrated in Figure 2 (see Appendix A.10 for a detailed walkthrough of a single reasoning cycle), which illustrates how the Routing Agent orchestrates distinct reasoning pathways in response to varying market conditions. For instance, in a clear, trending market (top panel), the agent can propose a streamlined and efficient workflow, such as `Search → Reasoning → Exit`, conserving computational resources. However, when faced with a more ambiguous or sideways market where signals may be conflicting (middle and bottom panels), the agent intelligently deepens the reasoning process. It initiates iterative loops involving the `Meta` workflow to scrutinize assumptions, resolve contradictions, and refine the analysis—as

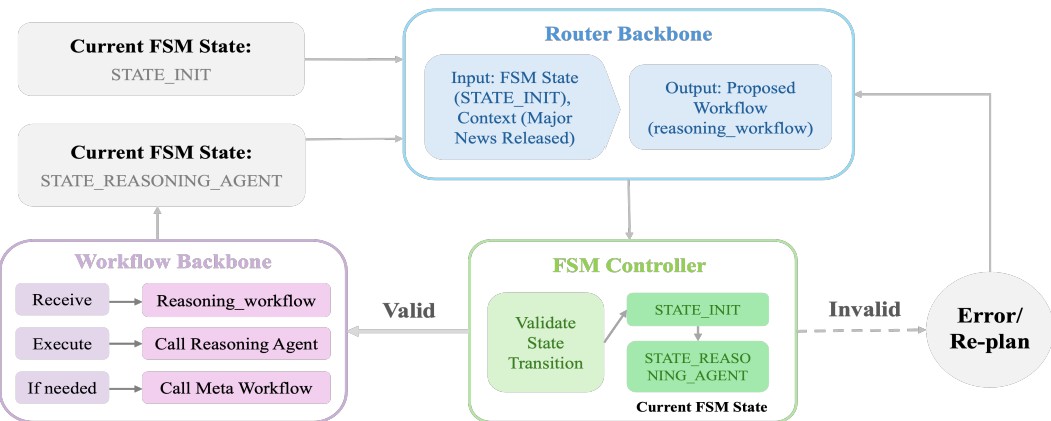

Figure 3: FSM Validation Process.

exemplified by the case where information retrieved during a `search` workflow contains contradictions, prompting a `Meta` review rather than directly proceeding to `Reasoning`. This adaptive mechanism ensures that the system's cognitive effort is dynamically calibrated to the complexity of the situation, enabling both efficiency in simple scenarios and robustness in complex ones.

**Validation - FSM Controller**: The Routing Agent's proposal is not immediately executed. Instead, it is submitted to the FSM Controller, which acts as the "rule enforcer". The FSM internally defines a set of legitimate state transition diagrams, explicitly outlining permissible workflow transitions and the exit conditions for various states. The FSM Controller then validates whether the proposed transition is legitimate. The primary objectives of this validation are twofold: (i) Ensuring Logical Coherence: It prevents illogical transition, such as selecting Reasoning Workflow without first conducting a Search Workflow. (ii) Preventing Infinite Loops: By setting constraints like maximum execution iterations, it mitigates the system endlessly oscillating between two states (e.g., selecting Search Workflow repeatedly), thereby ensuring the ultimate convergence of the entire inference process(see Figure 3).

**Execution - WorkflowBackbone**: Once a plan is approved by the FSM Controller, the WorkflowBackbone (the "execution unit") executes the corresponding workflow, allocating agents for the task. The output becomes the historical context for the next planning step, forming a closed-loop inference system. This tight coordination of planning, validation, and execution enables FinThink to adaptively decide whether to deepen its search (Search Workflow), reinforce reasoning (Reasoning Workflow), initiate refinement (Meta Workflow), or proceed to a decision (Exit Workflow).

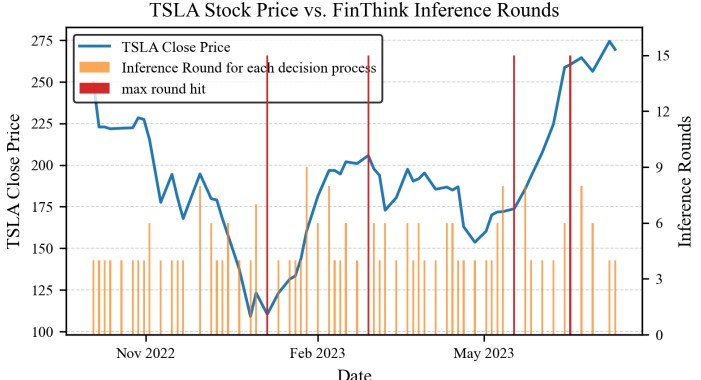

Figure 4: Interaction between TSLA Price Trajectories and Multi-Round FinThink Inference.

This structured adaptivity is empirically validated. As shown in Figure 4, the system's reasoning depth increases with market complexity: during stable periods it converges in few steps, while at critical inflection points—such as January, March, April and May 2023—it consistently reached maximum reasoning rounds. These episodes, marked by strong signal conflicts, highlight that CWRM is not a fixed linear process but a flexible framework that dynamically scales its cognitive effort to situational challenges.

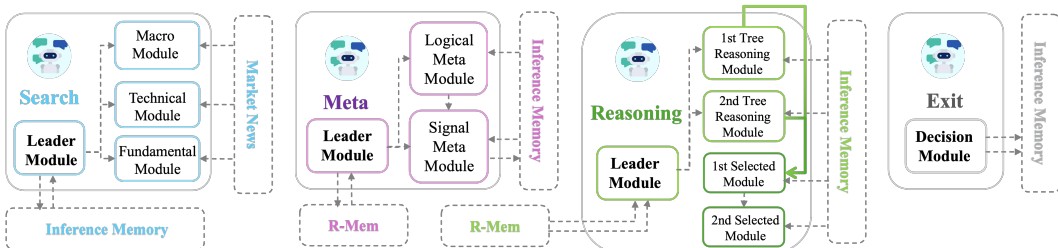

Figure 5: Overview of FinThink's Workflow Modules.

The WorkflowBackbone executes one of **four modular workflows**, the building blocks for the reasoning pipeline, as illustrated in Figure 5: **Search Workflow** acquires and synthesizes raw data; **Reasoning Workflow** forms the system's core analytical judgment; **Meta Workflow** acts as an internal quality inspector; and **Exit Workflow** transforms the final analysis into a trading action. Detailed descriptions are provided in Appendix A.7.

## 3.2 REASONING-DRIVEN HIERARCHICAL MEMORY SYSTEM (R-MEM)

One of the core advantages that differentiates FinThink from existing frameworks lies in its Reasoning-Driven Hierarchical Memory (R-Mem), which is designed in conjunction with the CWRM architecture to endow the system with a human-like capacity for reflective memory. In contrast, traditional LLM-based agents such as FinAgent (Zhang et al., 2024) and FinCon (Yu et al., 2024) typically rely on trigger-based or fragmented memory records, for example, passively invoking reflection only when PnL declines. Such memory lacks contextual completeness, preventing the formation of systematic cognition and meaningful lessons.

To achieve evolutionary adaptivity, R-Mem moves beyond reactive triggers and instead generates structured memories from deep reflection on entire trade cycles. Memories are stored in a structured vectorized format and retrieved using a Maximum Marginal Relevance (MMR) strategy to promote relevance and diversity. Critically, unlike prior work focusing on single-asset memory, we organize R-Mem by **Narrative Driver Clusters**—groups of assets sharing fundamental drivers.

This design is inspired by a crucial observation from real-world trading: expert traders rarely confine their knowledge to a single stock. Instead, they develop a "style" or expertise applicable to an entire class of assets (e.g., commodity sector stocks, large-cap tech stocks), allowing them to generalize experiences across their domain. This is particularly relevant under the AMH, where market-wide regime shifts mean that a lesson learned from one asset is often directly applicable to its peers. Our work represents a first attempt to operationalize this principle by partitioning memory based on assets' fundamental characteristics, enabling powerful cross-asset experiential transfer (e.g., applying insights from MSFT to GOOG) and drastically improving memory generalization.

The success of this approach is empirically validated through a qualitative analysis of the reflection memories.

As visualized in Figure 6, a semantic clustering of memories reveals that R-Mem develops specialized, context-aware reasoning patterns tailored to the unique dynamics of each asset cohort. What is particularly striking is that these distinct clusters emerged autonomously. The memory texts were vectorized without any explicit categorical labels, meaning the clear four-category separation is a result of the system's own emergent understanding of the underlying reasoning structures—a finding that we found highly compelling.

For the stable *Platform & Blue-Chip Cohort* (AAPL, GOOG, MSFT), memory clusters correspond to different external drivers, with keywords separating signals like 'macro' from 'fundamental'. This indicates a diversified and multi-faceted playbook. In stark contrast, for the volatile *High-Beta & Narrative-Driven Cohort* (TSLA, AMZN), the memory clusters reveal a playbook of trading tactics, dominated by keywords like 'oversold', 'short', and 'long'. At the same time, the *Platform & Blue-Chip Cohort* exhibits a dispersed geometric structure, while the *Narrative-Driven Cohort* forms distinctly compact geometries. This stark geometric contrast visually confirms that R-Mem has developed two fundamentally different cognitive models for the two asset classes.

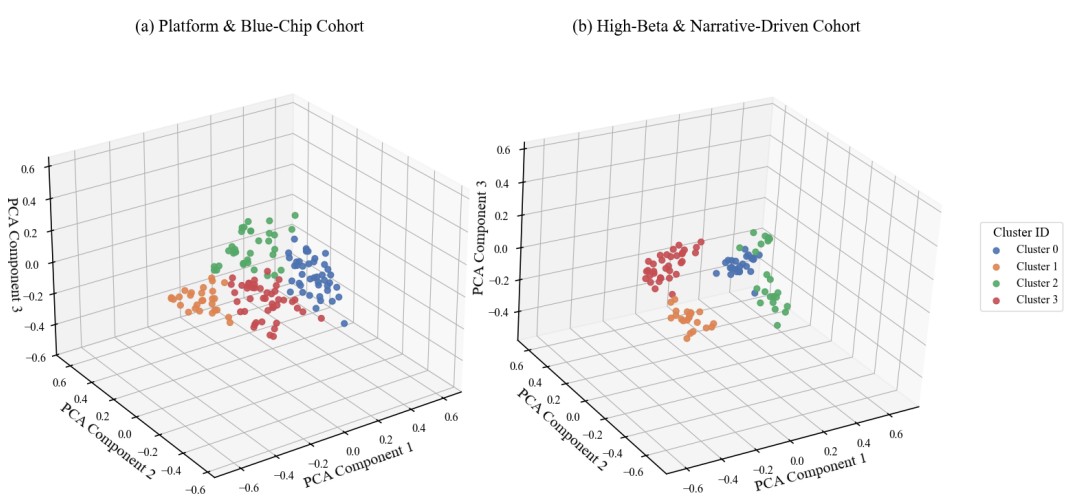

Figure 6: 3D visualization of semantic memory clustering for two distinct asset cohorts. The clusters emerged autonomously without any explicit labeling, revealing distinct reasoning patterns. **(a)** For the stable *Platform & Blue-Chip Cohort*, clusters are separated by external drivers (e.g., macro vs. fundamentals). **(b)** For the volatile *High-Beta & Narrative-Driven Cohort*, clusters reflect an action-oriented playbook (e.g., momentum and volatility management).

This emergent capability to adapt its reasoning methodologies, rather than just its knowledge base, is the core mechanism that engenders robust, nuanced decision-making and superior risk control. This is quantitatively demonstrated by our state-of-the-art results for the Sharpe Ratio, Calmar Ratio, and Maximum Drawdown in experiments.

## 3.3 MITIGATING REASONING COLLAPSE: COGNITIVE FUNCTION DECOUPLING VIA STL PROMPT PROTOCOL

While FinThink's CWRM handles dynamic scheduling, a deeper challenge in multi-agent systems is maintaining deep reasoning fidelity without falling into "Reasoning Collapse into Voting". This common failure mode occurs when unconstrained agents prematurely converge on conclusions, by-passing intermediate analytical steps and introducing biases (Liu et al., 2025b;a). To prevent this, we introduce the STL (Sentiment-To-Logic) Prompt Protocol, a cognitive scaffolding that ensures the quality and depth of FinThink's inference.

STL's core innovation is the strict decoupling of cognitive functions: analysis agents are constrained to generate objective scenario analyses, strictly prohibiting direct trading recommendations. This structural delineation of "analysis" from "decision-making" preserves logical integrity and hierarchical depth, crucial for robust multi-agent collaboration. Aligned with the Adaptive Market Hypothesis (AMH), STL systematically structures the Reasoning Workflow into key analytical pillars, as illustrated in Figure 7. This framework enables dynamic weight adjustment of attentional focus on different market drivers by dynamically adjusting the signal strength from macroeconomic, fundamental, and technical indicators based on market signal intensity, and facilitates proactive risk aversion when uncertainty is high. By embedding risk mechanisms directly into the reasoning process and preventing cognitive overload, STL ensures FinThink achieves high-quality, adaptive, and risk-controlled decision-making in complex financial markets. (For detailed mechanisms and examples, see Appendix A.8; for prompt templates, see Appendix A.9).

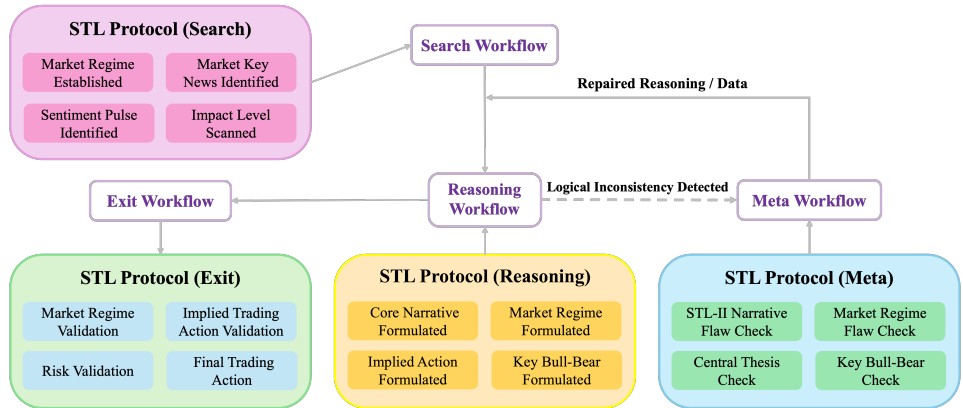

Figure 7: STL Protocol.

## 4 EXPERIMENTS

### 4.1 EXPERIMENTAL SETUP

We evaluate FinThink by backtesting its trading decisions on five major equities (AAPL, TSLA, MSFT, GOOG, AMZN) from October 5, 2022, to June 30, 2023. This specific period was deliberately chosen as it encapsulates a critical market cycle, transitioning from "rate hike panic" to a "market bottom", and finally to a "macro-driven rebound". This timeframe exhibits significant temporal and cross-sectional heterogeneity: some assets were highly sensitive to geopolitical events and interest rate shifts, while blue-chip stocks demonstrated greater resilience. Such diversity in market conditions and asset behaviors provides a robust environment for evaluating the model's stability and generalization capabilities. Our approach leverages a Reasoning-Driven Hierarchical Memory (R-Mem), pre-trained on data from June 1, 2022, to October 4, 2022, using a cross-asset learning strategy based on Narrative Driver Clusters, enabling the agent to generalize experiences across behaviorally similar assets. To mitigate the impact of random fluctuations, we report the median outcomes across five independent trials.

Importantly, all experiments are conducted with Gemini-2.0-flash, a lightweight and cost-optimized model, and thus fully reproducible via standard API access. By deliberately avoiding reliance on large, resource-intensive base models, we highlight that the observed gains stem from the architecture of FinThink itself, rather than the raw capacity of the underlying LLM.

For comparison, we benchmark against several recent multi-agent trading systems. We note that some state-of-the-art models, such as FinCon (Yu et al., 2024) and FinAgent (Zhang et al., 2024), present valuable methods but do not provide publicly available codebases. Furthermore, certain mechanisms, like FinCon's text-based gradient descent for memory enhancement, depend on highly customized experimental environments, making full replication challenging. Therefore, instead of attempting a strict baseline reproduction, we compare against their publicly reported metrics. To ensure a direct and fair comparison, their reported metrics are taken from the identical October 5, 2022, to June 30, 2023 backtesting period. A comprehensive description of our dataset, the cross-asset pre-training protocol, and simulation environment is provided in Appendix A.5.

### 4.2 EVALUATION METRICS

To ensure diagnostic clarity and directly assess our model's core reasoning and memory mechanisms, our evaluation focuses on a single-asset trading task. This approach avoids the confounding effects common in portfolio-level analysis, allowing for a rigorous inspection of the agent's decision-making process. We evaluate performance using four standard financial metrics: **Sharpe Ratio (SR)**: Quantifies risk-adjusted returns. **Total Return (TR)**: Measures overall profitability. **Maximum Drawdown (MDD)**: Assesses the largest peak-to-trough decline, indicating downside risk. **Calmar Ratio (CR)**: Evaluates the strategy's robustness by comparing return to its maximum drawdown. This selection of metrics provides a comprehensive view of both profitability and risk management. Detailed definitions and mathematical formulations are provided in Appendix A.4.

Table 1: Comprehensive performance comparison on AAPL, GOOG, and MSFT.

| Categories | Models | AAPL | | | | GOOG | | | | MSFT | | | |
|---|---|---|---|---|---|---|---|---|---|---|---|---|---|
| | | TR%↑ | SR↑ | MDD%↓ | CR↑ | TR%↑ | SR↑ | MDD%↓ | CR↑ | TR%↑ | SR↑ | MDD%↓ | CR↑ |
| Market | B&H | 22.315 | 1.107 | 20.659 | 1.080 | 22.420 | 0.891 | 21.191 | 1.058 | 27.856 | 1.230 | 15.010 | 1.856 |
| Our Model | FinThink | 11.640 | 1.630 | 3.050 | 3.816 | 23.840 | 1.620 | 5.910 | 4.034 | 17.800 | 1.680 | 8.060 | 2.208 |
| LLM-based | GA | 5.694 | 0.372 | 14.161 | 0.402 | -1.515 | -0.192 | 8.210 | -0.185 | -31.821 | -1.414 | 39.808 | -0.799 |
| | FINGPT | 20.321 | 1.161 | 16.759 | 1.213 | 0.242 | 0.011 | 26.984 | 0.009 | 21.535 | 1.315 | 16.503 | 1.305 |
| | FINMEN | 12.397 | 0.994 | 11.268 | 1.100 | 0.311 | 0.018 | 21.503 | 0.014 | -22.036 | -1.247 | 29.435 | -0.749 |
| | FINAGENT | 20.757 | 1.041 | 19.896 | 1.043 | -7.440 | -1.024 | 10.360 | -0.718 | -27.534 | -1.247 | 39.544 | -0.696 |
| | FINCON | 27.352 | 1.597 | 15.266 | 1.792 | 25.077 | 1.052 | 17.530 | 1.183 | 31.625 | 1.538 | 15.010 | 2.107 |
| DRL-based | A2C | 13.781 | 0.683 | 14.226 | 0.969 | 8.562 | 0.340 | 21.191 | 0.404 | 21.397 | 0.962 | 21.458 | 0.997 |
| | PPO | 14.041 | 0.704 | 22.785 | 0.616 | 2.434 | 0.097 | 25.202 | 0.097 | -4.761 | -0.214 | 30.950 | -0.154 |
| | DQN | 21.125 | 1.048 | 16.131 | 1.310 | 20.690 | 0.822 | 21.191 | 0.976 | 27.021 | 1.216 | 21.458 | 1.259 |

Table 2: Comprehensive performance comparison on TSLA and AMZN.

| Categories | Models | TSLA | | | | AMZN | | | |
|---|---|---|---|---|---|---|---|---|---|
| | | TR%↑ | SR↑ | MDD%↓ | CR↑ | TR%↑ | SR↑ | MDD%↓ | CR↑ |
| Market | B&H | 6.425 | 0.145 | 58.150 | 0.110 | 2.030 | 0.072 | 34.241 | 0.059 |
| Our Model | FinThink | 76.500 | 2.140 | 12.790 | 5.981 | 32.220 | 1.800 | 11.450 | 2.810 |
| LLM-based | GA | 16.535 | 0.391 | 54.131 | 0.305 | -5.631 | -0.199 | 37.213 | -0.151 |
| | FINGPT | 1.549 | 0.044 | 42.400 | 0.037 | -29.811 | -1.810 | 29.671 | -1.005 |
| | FINMEN | 34.624 | 1.552 | 15.674 | 2.209 | -18.011 | -0.773 | 36.825 | -0.489 |
| | FINAGENT | 11.960 | 0.271 | 55.734 | 0.215 | -24.588 | -1.493 | 33.074 | -0.743 |
| | FINCON | 82.871 | 1.972 | 29.727 | 2.788 | 24.848 | 0.904 | 25.889 | 0.960 |
| DRL-based | A2C | -35.644 | -0.805 | 61.502 | -0.580 | -12.560 | -0.444 | 37.106 | -0.338 |
| | PPO | 1.409 | 0.032 | 49.740 | 0.028 | 3.863 | 0.138 | 28.085 | 0.138 |
| | DQN | -1.296 | -0.029 | 58.150 | -0.022 | 11.171 | 0.398 | 31.174 | 0.358 |

Our primary objective is to maximize risk-adjusted returns, a more critical measure of a strategy's long-term viability than raw profitability. The experimental results strongly validate FinThink's success in this regard. Across all five tech stocks, FinThink consistently achieves the highest Sharpe Ratios and state-of-the-art Calmar Ratios, demonstrating superior performance where it matters most.

This outperformance is driven by its exceptional risk management, evidenced by dramatically lower Maximum Drawdowns (MDD). For example, on AAPL, FinThink reduces the MDD to a mere 3.050% from the B&H benchmark's 20.659%, while simultaneously boosting the Sharpe Ratio to 1.630 (vs. 1.107). While this disciplined, risk-first approach means FinThink does not always maximize Total Return (TR), it can still capture significant upside when market conditions are favorable. On TSLA, it delivered an impressive 76.500% TR—outperforming most benchmarks—paired with an exceptional Sharpe Ratio of 2.140 and a low MDD of 12.790%. In summary, FinThink successfully prioritizes stable, risk-adjusted growth over volatile, high-risk gains. This validates that its core design—the adaptive CWRM and reflective R-Mem—is highly effective at constructing robust strategies capable of navigating complex market dynamics.

## 5 CONCLUSION

In this paper, we introduced FinThink, a multi-agent system grounded in the Adaptive Markets Hypothesis. Its core innovations—a Context-aware Workflow for Reasoning (CWRM) for dynamic adaptation and a Reasoning-Driven Memory (R-Mem) that leverages Narrative Driver Clusters for cross-asset learning—enable superior risk-adjusted performance even with a lightweight LLM. Backtests confirm that FinThink significantly improves Sharpe Ratios while drastically reducing drawdown. While excelling at risk management, limitations include a trade-off between maximizing Sharpe Ratio and Total Return, and the need to validate our cross-asset memory mechanism in a full portfolio context. Future work will focus on extending FinThink to multi-asset portfolio optimization and developing more sophisticated memory fusion and retrieval algorithms for this complex setting. See Appendix A.11 for ethics and LLM disclosure.

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

## A APPENDIX

### A.1 ABLATION STUDY

To quantitatively assess the individual contributions of our core components, we conducted a rigorous ablation study. We focused on two pivotal innovations within FinThink: the Reasoning-Driven Hierarchical Memory (R-Mem) and the Sentiment-To-Logic (STL) Prompt Protocol. We systematically removed each component and evaluated the resulting model's performance on the volatile

TSLA dataset, which provides a challenging environment to test adaptability and risk management. The results are summarized in Table 3.

| Model Variant | TR (%) ↑ | SR ↑ | MDD (%) ↓ | CR ↑ |
|---|---|---|---|---|
| **FinThink (Full Model)** | **76.50** | **2.14** | **12.79** | **5.98** |
| w/o R-Mem | -17.49 | -0.99 | 25.36 | -0.68 |
| w/o STL Protocol | 13.60 | 1.61 | 6.44 | 2.11 |

Table 3: Ablation study results on the TSLA dataset. We compare the full FinThink model against variants with R-Mem and the STL protocol removed. TR, SR, and CR are "higher is better" (↑), while MDD is "lower is better" (↓).

The analysis of the ablation results reveals two critical insights:

**1. The Indispensable Role of R-Mem in Evolutionary Learning.** The most striking result is the catastrophic performance degradation upon removing the R-Mem module. The model's Total Return plummeted from 76.50% to -17.49%, and the Sharpe Ratio turned sharply negative (-0.99). This demonstrates that without the ability to reflect on entire trade cycles and distill structured, corrective heuristics, the agent is condemned to repeat its past errors. R-Mem provides the crucial mechanism for long-term, evolutionary adaptation, preventing the system from making the same costly mistakes in similar market conditions and thereby safeguarding capital.

**2. The Effectiveness of STL in Ensuring Deep Reasoning.** Removing the STL Prompt Protocol also led to a significant decline in performance, although the model remained marginally profitable. The Total Return dropped to 13.60%, and more importantly, the Sharpe Ratio fell drastically to 1.61, indicating a poor risk-return trade-off. This validates the STL's critical function in preventing "Reasoning Collapse". Without its structured, decoupled cognitive process, the multi-agent system's reasoning becomes shallow and susceptible to biases, failing to conduct the deep, logical analysis required to navigate market complexity and capture significant returns. The STL protocol is thus essential for maintaining the fidelity and depth of the reasoning process at each decision point.

In summary, this study confirms that FinThink's superior performance is not attributable to a single element but to the synergistic interplay of its core components. R-Mem enables the system to learn and evolve over time, while the STL protocol ensures the quality and robustness of its reasoning in the present moment. Both are vital for achieving stable, risk-adjusted returns in dynamic financial markets.

## A.2 QUALITATIVE ANALYSIS OF CUMULATIVE RETURN CURVES

In this section, we provide a qualitative analysis of the cumulative return curves for the five technology stocks in our backtesting period: TSLA, AAPL, AMZN, GOOG, and MSFT. These visualizations offer further insight into the behavioral patterns of the FinThink agent compared to other models.

A primary design philosophy of FinThink is the prioritization of risk-adjusted returns, with a specific focus on minimizing Maximum Drawdown (MDD) and maximizing the Sharpe Ratio (SR). This strategic choice means that the cumulative return (Total Return, TR) of FinThink may not always be the highest among all compared agents, particularly in periods of high market volatility where high-risk, high-reward strategies might temporarily outperform. As can be observed in Figures 8c, the FinThink curve (light blue) often appears less volatile and is typically positioned in the middle of the pack, demonstrating a consistent and stable appreciation of capital. This behavior underscores its ability to navigate uncertainty and avoid significant downturns, which is a critical aspect of long-term, sustainable trading strategies.

However, FinThink's adaptive nature allows it to capitalize decisively on clear market trends. For instance, in the backtests for TSLA (Figure 8d) and AMZN (Figure 8e), which exhibited strong directional trends during the evaluation period, FinThink's performance was notably more aggressive and highly profitable. When trading signals are clear and convergent, the agent acts with greater conviction, leading to superior returns that outperform most benchmarks.

Conversely, for large-cap blue-chip stocks such as MSFT (Figure 8c), GOOG (Figure 8a), and AAPL (Figure 8b), the market in Q4 2022 was characterized by range-bound, sideways movements. Under these ambiguous conditions, FinThink adopted a conservative, capital-preservation stance. It engaged in smaller-sized trades, prioritizing risk management while awaiting a clearer trend to emerge.

As the market direction became more apparent in Q1 2023, FinThink began to take more decisive positions. This strategy was exemplified in the trading of GOOG. During market oscillations, it effectively controlled risk, avoiding losses in uncertainty; and at the critical juncture of a trend reversal, it precisely seized opportunities, achieving significant returns. From the cumulative return curve, it is clear that FinThink initially performed steadily, then steadily climbed, gradually surpassing and ultimately outperforming almost all comparable models. This "defend first, then counter-attack" success highlights the exceptional adaptability and robustness of FinThink's strategy, demonstrating its advanced ability to dynamically switch between different market environments, being proficient in both offense and defense.

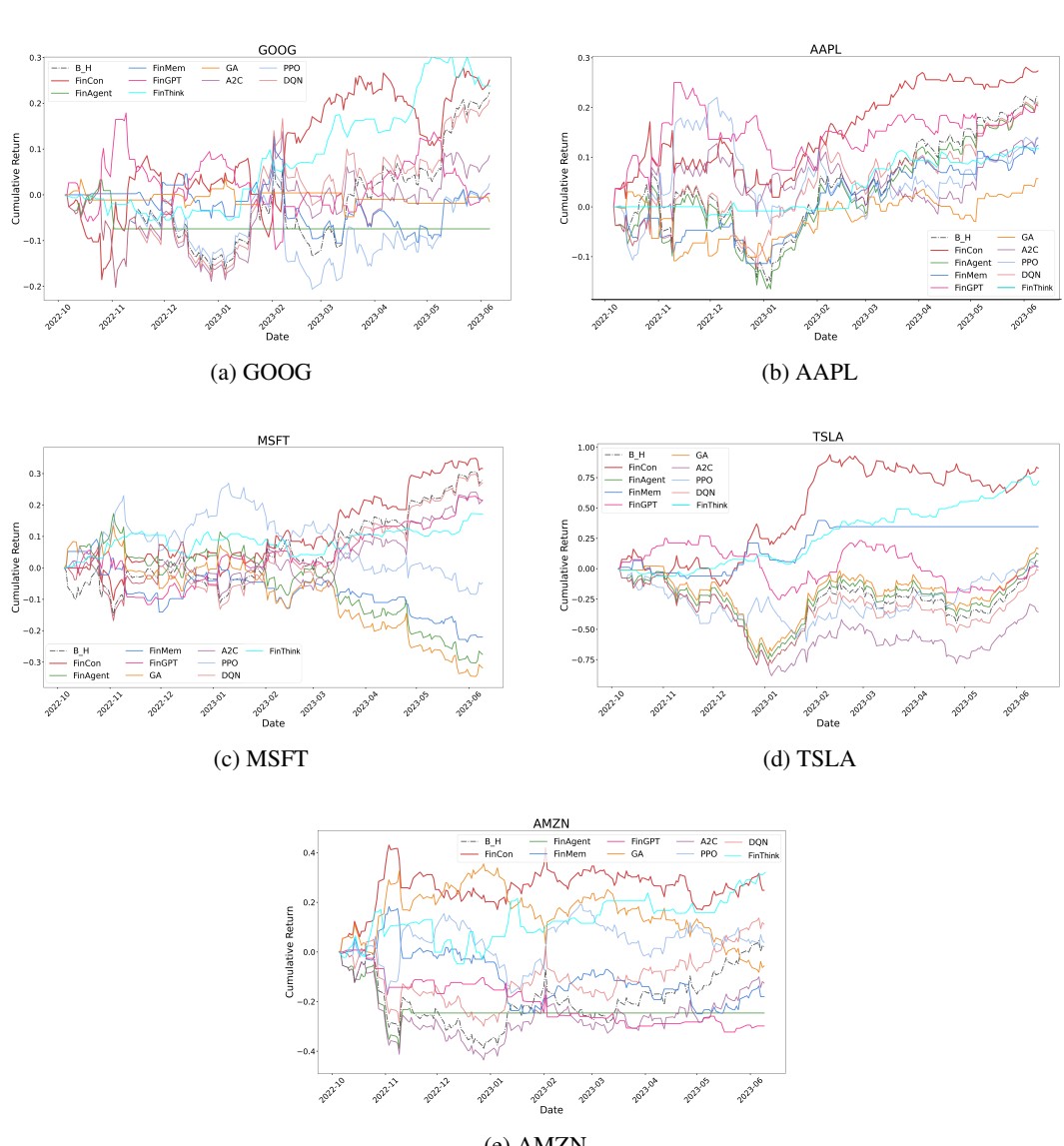

Figure 8: Cumulative Return comparison for GOOG, AAPL, MSFT, TSLA, and AMZN.

## A.3 CWRM PSEUDOCODE

---

**Algorithm 1** The FinThink Context-Aware Workflow for Reasoning (CWRM) Algorithm

---

**Global Components:**

*RoutingAgent*: An LLM-based planner that proposes the next workflow.

*FSMController*: A deterministic finite state machine that validates workflow transitions.

*WorkflowBackboneExecutor*: Executes specific workflows (Search, Reasoning, Meta, Exit).

*GlobalMemory*: A repository storing the outputs of all executed workflows.

**procedure** FINTHINK_TRADING_CYCLE($MarketInfo_t$, $Portfolio_t$)  ▷ Main procedure for a single trading decision at time step $t$.

    $fsm \leftarrow$ InitializeFSMController()

    $initialContext \leftarrow$ PrepareInitialContext(*MarketInfo_t*, *Portfolio_t*)

                ▷ Execute the core iterative inference loop to derive a final decision.

    $finalDecision, finalContext \leftarrow$ CWRM_INFERENCE_LOOP(*fsm, initialContext*)

             ▷ Execute the trade based on the final decision from the inference loop.

    $Portfolio_{t+1} \leftarrow$ ExecuteTrade(*Portfolio_t, finalDecision*)

    **return** $Portfolio_{t+1}$

**end procedure**

**procedure** CWRM_INFERENCE_LOOP(*fsm, initialContext*)

**Require:** $N_{max\_rounds}$: Maximum number of reasoning iterations.

    $historicalContext \leftarrow initialContext$

    **for** $i \leftarrow 1$ to $N_{max\_rounds}$ **do**

                                 ▷ *Phase 1: Planning*

        $proposedPlan \leftarrow$ RoutingAgent.planNextStep(*fsm.state, historicalContext*)

                              ▷ *Phase 2: Validation*

        **if** FSMController.validateTransition(*fsm.state, proposedPlan.goal*) **then**

            $fsm.state \leftarrow proposedPlan.goal$        ▷ Transition is valid, update state.

                            ▷ *Phase 3: Execution*

            $result \leftarrow$ WorkflowBackboneExecutor.execute(*proposedPlan*)

                        ▷ *Phase 4: Context Update*

            GlobalMemory.append(*result*)

            $historicalContext \leftarrow$ GlobalMemory.getRecentHistory()

            **if** $proposedPlan.goal$ = "Exit" **then**

                $decision \leftarrow$ ParseDecision(*result*)

                **return** *decision, historicalContext*       ▷ Terminal state reached.

            **end if**

        **else**

            log("Invalid transition proposed by RoutingAgent")

            **break**                          ▷ Loop is terminated.

        **end if**

    **end for**

    **return** "HOLD", *historicalContext*       ▷ Default action if max rounds reached.

**end procedure**

---

## A.4 FORMULA

To ensure diagnostic clarity and directly assess FinThink's core reasoning and memory mechanisms, our evaluation focuses on a single-asset trading task. This approach avoids the confounding effects common in portfolio-level analysis, allowing for a rigorous inspection of the agent's decision-making process. We evaluate performance using four standard financial metrics. For baseline comparison, we adopt the reported and reproducible results from FinCon (Yu et al., 2024) , as it is the only system that provides evaluation over the same backtesting period (October 5, 2022 to June 30, 2023).

(i) **Total Return (TR)**: This metric calculates the percentage change in the portfolio's value over the entire backtesting period.

$$\text{TR} = \frac{\text{Ending Value} - \text{Beginning Value}}{\text{Beginning Value}} \times 100\%$$

(ii) **Sharpe Ratio (SR)**: Measures the risk-adjusted return, quantifying the excess return per unit of risk. A higher Sharpe Ratio indicates a better return for the amount of risk taken.

$$\text{SR} = \frac{E[R_p - R_f]}{\sigma_p}$$

(iii) **Calmar Ratio (CR)**: Evaluates the risk-reward profile by comparing the average annual return to the maximum drawdown, assessing the strategy's robustness. Given that our back-testing period is less than one year, we approximate the Calmar Ratio using the total return and maximum drawdown over the 9-month period. Notably, while FinCon (Yu et al., 2024) did not report Calmar Ratio in their paper, their reported return and maximum drawdown over the identical backtesting window permit a consistent computation of this metric. In both cases, the Calmar Ratio is derived by dividing the 9-month total return by the corresponding 9-month maximum drawdown, ensuring methodological parity and a strictly fair comparison.

$$\text{CR} = \frac{\text{Total Return (over 9 months)}}{\text{Maximum Drawdown (over 9 months)}}$$

(iv) **Maximum Drawdown (MDD)**: Measures the largest peak-to-trough decline in account value during a specified period, reflecting the system's potential downside risk. A lower MDD indicates greater portfolio stability.

$$\text{MDD} = \max_{t_1 < t_2} \frac{\text{Portfolio Value}(t_1) - \text{Portfolio Value}(t_2)}{\text{Portfolio Value}(t_1)} \times 100\%$$

## A.5 DETAILED EXPERIMENTAL SETUP

### A.5.1 DATASET AND SIMULATION ENVIRONMENT

Our evaluation spans the period from June 1, 2022, to June 30, 2023, and is explicitly divided into two phases: a *memory training period* (June 1, 2022 – October 4, 2022) used for constructing and calibrating the reflection memory, and a subsequent *real backtesting period* (October 5, 2022 – June 30, 2023) used for performance evaluation. This design ensures that memory construction and evaluation are temporally separated, thereby preventing information leakage across phases. All trading decisions are simulated using daily closing prices and are conditioned strictly on news available up to that day to prevent data snooping. To maintain methodological consistency with the FinCon (Yu et al., 2024) benchmark, our simulation permits both long and short positions, thereby aligning our experimental setup with prior work and ensuring direct comparability of results.

Equity news data are obtained from public financial feeds such as **Alpaca and Benzinga**, while macroeconomic releases including CPI, Federal Reserve announcements, and unemployment reports are sourced directly from **Federal Reserve's official repository**.

To ensure statistical reliability and reflect real-world conditions, we report the median outcome across 5 independent trials. For decision generation, we set the temperature parameter to **0.3** to balance determinism and flexibility. No random seed configuration is required. Table 4 summarizes the data composition.

### A.5.2 CROSS-ASSET LEARNING PROTOCOL VIA NARRATIVE DRIVER CLUSTERS

A core limitation of traditional agent memory systems in finance is their inability to generalize learned experiences across diverse assets. Our Reasoning-Driven Hierarchical Memory (R-Mem) addresses this gap by introducing *Narrative Driver Clusters*. This paradigm organizes memories based on underlying economic and behavioral factors driving asset prices, rather than by individual ticker symbols, to enable cross-asset experiential transfer.

Table 4: Summary of News Articles and Macroeconomic Data.

| Category | Count |
|---|---|
| **Equity-Specific News (May 2022 - Jun 2023)** | |
| Apple (AAPL) | 4,916 |
| Amazon (AMZN) | 3,696 |
| Google (GOOG) | 2,439 |
| Microsoft (MSFT) | 2,930 |
| Tesla (TSLA) | 6,831 |
| **Macroeconomic Signals (May 2022 - Jun 2023)** | |
| CPI Releases | 13 |
| Federal Reserve Announcements | 10 |
| Unemployment Reports | 16 |

Based on the distinct characteristics of the five assets in our experiments, we partitioned the memory repository into two logically cohesive cohorts:

- **High-Beta & Narrative-Driven Cohort:** Includes Tesla (TSLA) and Amazon (AMZN), which share heightened sensitivity to macroeconomic liquidity and compelling industry narratives. Memories focus on navigating high-volatility regimes.

- **Platform Ecosystem & Blue-Chip Cohort:** Comprises Microsoft (MSFT), Apple (AAPL), and Google (GOOG), characterized by large market caps and resilient business models. Memories concentrate on long-term value assessment and defensive strategies.

To validate this approach, we designed a specific "warm-up" protocol involving pre-training and backtesting.

*Memory Pre-training Phase

- **Period:** To ensure the rigor of our evaluation, our memory training was strictly confined to the period from June 1, 2022, to October 4, 2022. This deliberate temporal separation ensures that the model cannot access any future market information during the subsequent live backtesting period (after October 5, 2022), thereby effectively preventing implicit look-ahead bias.

- **Process:** The agent operated in a simulated environment, iterating through the historical data for 3 epochs. After each simulated trade cycle for an asset, a reflection memory was generated and stored in its corresponding cluster, building a foundational set of structured memories. This process, specifically, is what we refer to as the memory pre-training phase.

- **Training Configuration:** The **High-Beta & Narrative-Driven Cohort** was trained on a mixed dataset of TSLA and AMZN. The **Platform Ecosystem & Blue-Chip Cohort** was trained on a concentrated dataset of MSFT, AAPL, and GOOG.

*Backtesting Phase

- **Process:** Following pre-training, we conducted separate backtests for each of the five assets over the main evaluation period (October 5, 2022, to June 30, 2023).

- **Cohort-Specific Application:** Crucially, assets only leveraged the R-Mem instance pre-trained on their own cohort. For example, the backtest for TSLA utilized the R-Mem instance pre-trained on the mixed TSLA/AMZN dataset, as did the backtest for AMZN.

This experimental design directly tests our central hypothesis regarding cross-asset generalization. By co-training on clustered assets, we compel R-Mem to learn generalizable heuristics rather than asset-specific idiosyncrasies. For instance, a lesson about misinterpreting momentum signals during a market-wide liquidity contraction, learned from a TSLA trade, becomes immediately available to inform a subsequent trading decision on AMZN under similar conditions.

### A.5.3 FURTHER DISCUSSION ON NARRATIVE DRIVER CLUSTERS

As stated in the main text, our partitioning of R-Mem into Narrative Driver Clusters is fundamentally a heuristic design, inspired by observations of real-world traders and our interpretation of the Adaptive Markets Hypothesis (AMH). We acknowledge this approach is rooted in human-centric observation, but we argue that its significance lies in this very characteristic. It represents a pioneering attempt to structure an agent's memory in a manner analogous to human experts—by style, sector, or shared economic drivers—rather than by isolated ticker symbols, a path previously untrodden in this domain.

Intuitively, this paradigm is both scalable to a larger number of assets and extensible to more granular categories. The trading behavior of human experts provides compelling real-world evidence for this. For example, a trader active in the oil and gas industry does not confine their focus to a single stock. Instead, their portfolio often encompasses numerous companies across the upstream, midstream, and downstream segments of the value chain. Their ability to navigate this complex ecosystem effectively would be difficult to explain without some form of consensus experience transfer, where a lesson learned from one asset is generalized and applied to its peers under similar market conditions.

Due to space limitations, a full exploration of this concept could not be undertaken in the present work. However, we believe this is a promising direction for future research, which could investigate more dynamic or data-driven methods for defining and evolving these memory clusters.

### A.5.4 R-MEM VIA MMR

To ensure that the agent can effectively leverage its accumulated experiences, memories generated by R-Mem are stored in a structured vectorized format. We then implement a sophisticated two-stage retrieval mechanism based on **Maximum Marginal Relevance (MMR)** to promote both relevance and diversity in the retrieved information. Unlike prior work focusing on single-asset memory, our approach partitions memory based on assets' fundamental characteristics to enable cross-asset transfer. This process is initiated when a **meta leader** or **reasoning leader** generates a query instruction to search for relevant past memories. The goal is to retrieve memories that are not only highly relevant to the current market context but also sufficiently diverse to prevent cognitive fixation and encourage robust decision-making.

The retrieval process is structured as follows:

1. **Coarse-grained Recall:** This initial stage is designed for rapid, broad filtering. The query instruction generated by the leader is matched against the `reflection_title` of all active memories using a high-performance FAISS (Facebook AI Similarity Search) index. This step efficiently identifies a candidate pool of the top-30 most relevant memories. If the total number of active memories is less than 30, all available memories are included in the candidate pool.

2. **Fine-grained Reranking and Diversification:** The candidate pool from the first stage undergoes a more precise reranking using the MMR algorithm. In this phase, a detailed relevance score is computed by combining two similarity metrics: (1) the similarity between the query title and the memory's `reflection_title`, and (2) the similarity between the query's descriptive component and the memory's `assessment_description`. These two scores are weighted equally ($\alpha = 0.5$) to form a final relevance score. Finally, the top-3 memories after MMR reranking are selected and returned to the workflow.

   Subsequently, the MMR algorithm iteratively selects the top-3 memories from the candidate pool. The selection criterion balances relevance with diversity, governed by the formula where relevance is weighted by $\lambda = 0.7$ and a diversity penalty (calculated as the maximum similarity to already selected memories) is weighted by $1 - \lambda = 0.3$. This configuration ensures that the final retrieved memories are both highly applicable and varied, providing a rich set of heuristics for the agent's subsequent reasoning steps.

### A.6 MARKET COGNITION MODULE

The Market Cognition module acts as a crucial bridge between raw market information and the sophisticated reasoning processes of FinThink. Specifically, the **Market Cognition** module processes

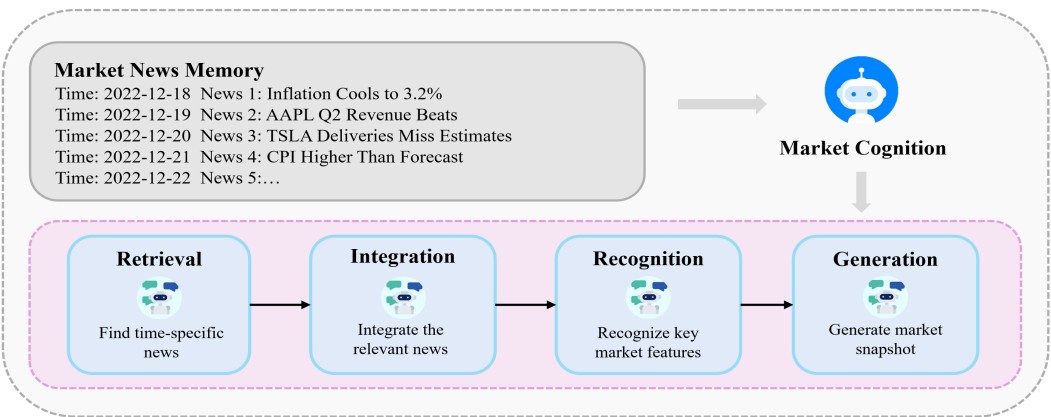

Figure 9: Market Cognition

the key macroeconomic, fundamental, and news data extracted by the Search Workflow. Every 14 days, it synthesizes this information into a comprehensive market summary, which is then provided to the Reasoning Workflow for in-depth analysis. This periodic summarization ensures that the agent's analytical foundation is consistently updated with a stable, long-term market worldview, rather than being overly influenced by transient daily noise. This structured aggregation of market context is vital for maintaining the depth and breadth of the subsequent reasoning and decision-making processes.

### A.7 DETAIL OF WORKFLOW

CWRM decomposes financial analysis tasks into four highly cohesive core workflows, which collectively form a complete cognitive pipeline from raw information to final trading decisions.

**Search Workflow**: This workflow acquires necessary data (macroeconomic, technical, fundamental) from the external environment. Critically, it leverages the MarketCognitionTracker Module (see Appendix Figure 9) to synthesize market reports, interpreting short-term signals within a long-term market worldview to provide multi-scale context for subsequent stages.

**Reasoning Workflow**: As the core analytical engine, this workflow uses specialized agents to deduce and integrate the collected data into a structured report. The report crystallizes the system's judgment into four key components: Market Regime, Core Narrative, Key Conflict, and Alignment Check.

**Meta Workflow**: Functioning as an internal quality inspector, this workflow scrutinizes prior inferences for logical flaws, unverified assumptions, or contradictions. Upon detecting deficiencies, the Routing Agent initiates corrective actions by re-invoking the Search or Reasoning workflows, ensuring the inference process continually converges toward higher-quality conclusions.

**Exit Workflow**: As the final inference stage, this workflow transforms the consolidated analytical report into an executable action. It synthesizes the Narrative strength, Market Regime robustness, and signal validation to compute a confidence level (High, Medium, Low, N/A). Based on this level, it determines the final action (e.g., BUY, SELL, HOLD) and position size, outputting a transparent, traceable JSON instruction.

### A.8 DETAILED DESCRIPTION OF THE STL (SENTIMENT-TO-LOGIC) PROMPT PROTOCOL

A critical challenge in multi-agent systems is "Reasoning Collapse into Voting", a failure mode where hierarchical reasoning degrades into a superficial chain-of-voting. FinThink mitigates this through the STL (Sentiment-To-Logic) Prompt Protocol, which enforces a strict decoupling of analytical and decision-making functions. The impact of this protocol is starkly illustrated in Figure 10. As shown in the right panel, a system without STL is paralyzed by conflicting market signals. The reasoning process collapses into a simple voting mechanism (BUY vs. SELL), resulting in a

Reasoning Workflow's Output：
- Market Analysis:Bearish continuation.
- STL-II:Leaning SHORT. Bearish trend continues, and heuristic memory (uuid=550e8400-e29b...) supports ignoring short-term rebounds during such phases.

Meta Workflow's Output：
- Flaw detection:Prior analysis overemphasized the bearish trend.
- Justification: Need to examine the conflict between oversold signals and bearish continuation. Heuristic memory (uuid=6fa459ea-ee8a...) shows past losses from ignoring rebound signals.

Reasoning Workflow's Output：
- Market Analysis:Bearish.
- STL-II:Leaning LONG. Meta highlighted a strong short-term rebound, validated as reasonable. Heuristic memory (uuid=123e4567-e89b...) also supports oversold rebound scenarios.

Exit Workflow's Output：
- **Action: Transfer to BUY .**
- Reasoning: Classified as a Justified Constraint. While the market remains broadly bearish, prolonged oversold conditions and rebound potential justify a small speculative LONG position.

**w/ STL: Structured Reasoning and Corrective Refinement**

Reasoning Workflow's "**Vote**"
- Action: Proposes BUY 700 Shares.
- Justification (Simplified): "The technical charts show a bullish crossover. Momentum is positive. This is a clear entry point for a long position."

Meta Workflow's "**Counter-Vote**"
- Action: Proposes SELL 1000 Shares.
- Justification (Simplified): "This ignores the terrible macroeconomic news. The Fed is signaling another rate hike. The market is risk-off. This is an obvious short."

Reasoning Workflow's "**Vote**"
- Action: Proposes HOLD.
- Justification (Simplified): "Since the previous analyses are contradictory, I believe the current market is in conflict; therefore, no buying or selling should be executed."

3. Exit Workflow's Final Action
- **Action: Defaults to HOLD (paralyzed state).**
- Reasoning: "Critical conflict detected. The system has provided contradictory BUY and SELL recommendations with equal conviction. Unable to form a coherent strategy. To avoid erratic behavior, no new action will be taken."

**w/o STL:  Reasoning Collapse into Voting**

Figure 10: STL vs. Non-STL Reasoning Outcomes — Structured Refinement vs. Collapse into Voting

HOLD state born from indecision, not strategy. This cognitive paralysis demonstrates a failure to learn, as no heuristic memory is referenced. In stark contrast, the left panel shows how the STL protocol transforms this conflict into a productive refinement loop. The Meta Workflow identifies a flaw in the initial analysis by explicitly citing heuristic memory of past errors. This triggers a second reasoning cycle that re-evaluates the evidence and produces a confident, well-justified contrarian decision. By enforcing structure and leveraging memory, STL prevents system failure and enables robust, adaptive reasoning.

## A.9    AGENT PROMPT TEMPLATES

This appendix provides the detailed prompt templates that steer the core agents within the FinThink framework. These templates enforce structured reasoning, mandate the use of historical heuristics, and decouple analytical tasks from decision-making to prevent cognitive collapse.

### A.9.1    FINAL REASONING SYNTHESIS AGENT

This agent performs the final comprehensive synthesis based on multi-source information to form a structured, unbiased market judgment. Its output serves as the core basis for subsequent decision-making.

**Prompt: Final Reasoning Synthesis**
- Overall Goal: *[Goal Description]*
- Task Context: *[Question/Problem Statement for the Asset]*
- Current Portfolio: *[Portfolio Summary]*
- Agent Role: Final Synthesizer
- Input Context: *[Chain history from upstream agents]*
- Memory Heuristics: *[List of retrieved corrective heuristics]*

**Core Task: Comprehensive Synthesis for *[Target Symbol]***

Based strictly on the provided input context and historical wisdom (corrective heuristics), synthesize a refined and objective analysis. Your output must be purely analytical and **MUST NOT** provide a direct BUY, SELL, or HOLD recommendation.

**Heuristic Usage & Inline Citation Rule (Tiered Approach)**

You must support your reasoning with historical wisdom when applicable, using one of the following citation formats.

1. **Tier 1: Contextual Citation** (For high-level warnings or pattern recognition): Cite when a memory's theme provides crucial context, but its specific heuristic is not directly applicable.

2. **Tier 2: Prescriptive Citation** (For direct application of corrective actions): Cite when a memory's heuristic is directly and logically applicable to justify your reasoning (e.g., adjusting signal weights).

**Analysis Steps & Output Structure (Strictly Adhere)**

1. **Best Narrative Construction**
   - **Best Narrative Thesis:** A concise summary of the core narrative (max 50 words).
   - **Causal Narrative Synthesis:** A detailed paragraph synthesizing how macro, fundamental, sentiment, and technical factors interact.
   - **Structured Narrative Components:**
     – SentimentPulse: [Strongly Bullish — ... — Strongly Bearish]
     – ImpactLevel: [High — Moderate — Low]
     – Expected Positioning Response: A description of anticipated market behavior.
     – Observed Technical Confirmation: Bullet points on Price Action, Momentum, Volume/Flow.
     – Factual Anchor Points: 2-3 critical facts from Search findings underpinning the narrative.

2. **Market Regime Classification**
   - Classification Result: [Strong Bullish — ... — Strong Bearish]
   - Regime Polity: A one-sentence characterization of the regime's nature.
   - Regime Synthesis Explanation: Explain how macro, technical, and fundamental signals were integrated and reconciled to derive the classification.

3. **Overall Synthesis & Assessment:** Summarize the current state and dominant narrative/bias.

4. **Key Bullish Factors (Evidence-Based):** List the 2-3 most significant positive factors with their evidence source and weighted significance.

5. **Key Bearish Factors (Evidence-Based):** List the 2-3 most significant negative factors with their evidence source and weighted significance.

6. **Significant Conflicting Factors & Uncertainty Assessment:** Identify high-weight conflicting signals, explain the nature of the conflict, and describe the resulting uncertainty.

7. **Risk/Reward Profile – STL-II: From Sentiment to Dynamics**
   - 7.1. Structural Alignment Check: Compare the implied behavior from the Best Narrative with the Market Regime. *Outcome: Aligned or Misaligned.*
   - 7.2. Structure-Violation Justification (Only if Misaligned): Determine if the misalignment is a calculated tactical opportunity or an unsupportable violation. *Outcome: Justified Violation, Unjustified Violation, or N/A.*
   - 7.3. Sentiment–Outcome Transition (STL-II Path Analysis): Determine the reasoning path's integrity. *Outcome: Robust (Structural), Vulnerable (Tactical), or Broken.*
   - 7.4. Final Qualitative Risk Synthesis: Classify the final risk posture. *Choice: High-Conviction Action — ... — Conflicted/Neutral Stance.*

8. **Timescale Considerations:** Identify and discuss any divergence between short-term and long-term signals and its implications.

### A.9.2 FINAL META AGENT (DIRECTIVE FORMULATION)

This agent acts as a Strategic Review Officer, responsible for reviewing defects found in the previous reasoning cycle. Its task is to generate a clear, actionable 'Analytical Directive' for the next reasoning agent based on historical experience and the current snapshot.

**Prompt: Meta Directive**
- **Overall Goal:** Formulate an Analytical Directive to correct a reasoning flaw.
- **Primary Input:** *[Defect Analysis Report from the initial Meta-agent]*
- **Reference Input 1:** *[Original Reasoning Output and Search Data Snapshot]*
- **Reference Input 2:** *[Historical Wisdom/Reflections linked to cited Memory IDs]*
- **Agent Role:** Strategic Review Officer

---

**Guiding Philosophy & Core Constraints**

Your goal is to issue a clear, actionable **Analytical Directive** for the next reasoning cycle, correcting the paradigm flaw identified in the defect report.

1. **Critical Instruction:** You must actively integrate the lessons from *[Historical Wisdom]* into your directive in natural language, demonstrating a deep understanding of the parallel between the past situation and the current one. Do not explicitly cite UUIDs in the final output.

2. **The Snapshot Principle:** All directives must be actionable based *only* on currently known information. Do not formulate plans requiring future observation.

**Task: Generate a Structured, Actionable Directive**

1. **Central Thesis for the Next Reasoning Cycle**
   - **Thesis Statement:** A single, concise, falsifiable question or conflict statement.
   - **Origin & Historical Link:** Explain why this thesis is critical by connecting the current situation to a historical parallel provided in the reflections.
   - **Causal Context:** Indicate where this thesis sits within the causal chain (e.g., structural bias).
   - **Task Allocation Intent:** Specify what the downstream agent must investigate based on this thesis.
   - **Risk-Opportunity Anchor:** Define the primary risk if the thesis holds true and the potential opportunity if it is falsified.

2. **Key Conflicting Evidence from Current Snapshot**
   - **Evidence For Side A:** Specific data point(s) supporting one side of the conflict.
   - **Evidence For Side B:** Opposing data point(s), especially what was previously missed.

3. **Proposed Analytical Adjustment (Heuristic)**
   - **The Rule:** Propose a clear rule or focus shift for the next reasoning cycle, directly inspired by a historical lesson.
   - **Justification from Historical Wisdom:** Explain why this adjustment is the correct response. State the principle learned from a past event and argue for its applicability now by comparing the historical context with the current snapshot.
   - **Constraint: The Snapshot Principle in Action**
     Your heuristic must be a rule for interpreting **current, known data**. It must not be a plan requiring future data.
     - Allowed Heuristics (Current Snapshot Interpretation):
       * *"Given elevated macro uncertainty in the current snapshot, downgrade technical signal weights by one tier for this cycle."*
       * *"In a confirmed structural breakdown (per current data), de-weight short-term oversold oscillators as noise."*
     - Forbidden Heuristics (Requires Future Observation):...

### A.9.3 DECISION PROTOCOL AGENT

This agent is responsible for synthesizing the final analytical outputs into a quantifiable trading action. It follows a strict protocol to ensure decisions are logical, risk-managed, and consistent with the framework's principles.

**Objective** This protocol aims to make trading decisions by establishing the strategic direction via Market Regime and integrating the STL-II action validation module as a decisive tactical filter. STL-II is used not only to assess the reasonableness of actions but also possesses robust risk control capabilities, thereby enabling flexible trading and effective risk management. In cases where the signals from the Market Regime and STL-II are inconsistent or misaligned, this protocol shall prioritize **Cover** over **Hold**.

**Decision Terminology & Confidence Levels**
• **Decision Action:** BUY, SELL, HOLD, COVER LONG, COVER SHORT
• **Confidence Level:** High, Medium, Low, Very Low

---

**Guiding Principles**

1. **Alignment Principle:**
   Complete consistency between the directional signals of the Market Regime and STL-II is the primary prerequisite for initiating a trade. This signifies a high-quality signal. In this context, opening a position is encouraged; if a position already exists and the risk assessment is appropriate, adding to the position is permissible.

2. **Conflict Principle:**
   Any directional inconsistency (misalignment) between the Market Regime and STL-II, unless explicitly labeled as "Justified Contrarian," should be treated as a strong risk warning signal. The following rules apply:
   - **If a position contrary to the regime's direction is held:** It should be immediately covered.
   - **If a position aligned with the regime's direction is held:** Reducing or covering the position should be considered.
   - **If no position is held:** Opening a new position is strictly prohibited.

3. **Exception Principle:**
   A signal explicitly labeled by STL-II as "Justified Contrarian" is the only scenario where opening a position is allowed amidst conflicting signals. Such trades are classified as high-risk and must utilize a **"Very Low"** confidence level, with the position opened strictly according to the direction indicated by STL-II.

**Decision Process**

1. **Extract Core Inputs**
   - Regime Direction
   - STL-II Direction
   - Qualitative Tag (e.g., "Justified Contrarian")
   - Current Position

2. **Evaluate Signal Consistency (Alignment Check)**
   - **Scenario A: Alignment**
     The Regime Direction is perfectly consistent with the STL-II Direction. This is a high-quality trading signal.
     – If no position is held or an aligned position exists, prepare to BUY/SELL or add to the position. Proceed to Step 3.
     – If a contrary position is held, immediately cover and consider a reverse position ("cover and reverse").

- **Scenario B: Misalignment**
  The Regime Direction is inconsistent with the STL-II Direction.
  - **Check for Exception:** Is the STL-II Qualitative Tag "Justified Contrarian"?
    * **Yes (Exception applies):** Prepare to open a position with **Very Low** confidence, strictly following the STL-II direction. Proceed to Step 4.
    * **No (Dangerous signal):** If any position is held, execute **COVER LONG** or **COVER SHORT**. Otherwise, strictly **HOLD**. The process ends.

3. **Assign Confidence Level for "Aligned Signals"**
   This step applies only to "Alignment" signals from Step 2.

   - **Base Confidence:** A Strong Regime sets a *High* base confidence; a Weak Regime sets a *Medium* base confidence.
   - **Confidence Adjustment:**
     - If the STL-II tag is *Aligned and Sustainable* or *High-Conviction*, elevate the confidence level (e.g., Medium $\rightarrow$ High).
     - If the STL-II tag is *Aligned but Cautious*, the confidence level remains unchanged.
     - If a comprehensive assessment reveals significant concerns, downgrade the confidence level (e.g., Medium $\rightarrow$ Low).
     - If the STL-II tag is *Structurally Unsound*, this has veto power; the decision is downgraded to **HOLD**.

4. **Calculate Final Quantity**

   - **Base Quantity:** Assume Total Assets = \$1,000,000. The **Base Quantity** is defined as 40% of total assets, representing \$400,000 worth of stock.
   - **Confidence Multiplier:** High: 1.0, Medium: 0.8, Low: 0.5, Very Low: 0.2.
   - **Signal Multiplier:** For BUY/SELL/COVER, the multiplier is 1.0; for HOLD, it is 0.
   - **Final Formula:**

     $$\text{Final Quantity} = \text{Base Quantity} \times \text{Signal Multiplier} \times \text{Confidence Multiplier}$$

5. **Generate Final Output (JSON Only)**

   ```
   {
     "Reasoning":  "<Within 400 words.  Must clearly
   explain the decision process:  1.  State Inputs:  Re-
   port the Market Regime and the STL-II analysis re-
   sult.  2.  Declare Logical State:  State 'Alignment'
   or 'Misalignment'.  3.  Trace Reasoning Link:  Ex-
   plain how the inputs map to the final decision by
   referencing the triggered protocol principle.  4.
   State Final Action:  Clearly state the final 'Deci-
   sion' and 'Confidence'.>",
     "Decision":  "<BUY | SELL | HOLD | COVER LONG |
   COVER SHORT>",
     "Quantity":  "<Integer>",
     "Confidence":  "<High | Medium | Low | Very Low |
   N/A>"
   }
   ```

## A.10 DETAILED WALKTHROUGH OF A SINGLE REASONING CYCLE (GOOG, 2023-02-06)

This appendix provides a step-by-step breakdown of the FinThink system's reasoning process for a single trading decision concerning Google (GOOG) on February 6, 2023. It showcases the iterative loop of information gathering (Search), analysis (Reasoning), self-critique (Meta), and final action (Exit), demonstrating the system's dynamic and reflective capabilities.

### Step 1: Search Workflow

**Agent Role:** Lead Search Coordinator
**Objective:** Synthesize raw market data into structured, sentiment-driven signal chains.

1. DETAILED FACTUAL LIST
- **Macro:** US Unemployment Rate drops to 3.4% (stronger economy). Fed raises rates by 25bps.
- **Technical:** GOOG at $105.20, above key SMAs/EMAs. MACD is positive. Overall technical momentum is strong, but Stochastic Oscillator is overbought.
- **Fundamental:** YouTube ad revenue $7.96B. GOOG shares declined post-earnings due to AI competition fears (Microsoft/ChatGPT). Google is testing its own AI chatbot and conducting layoffs.

2. SENTIMENT-DRIVEN SIGNAL CHAINS (STL)
- **Chain 1 (Bullish Macro):** Resilient labor market reduces recession fears, potentially leading to a market rally.
- **Chain 2 (Bearish Fundamental):** AI competition from Microsoft threatens Google's search dominance.

### Step 2: Initial Reasoning

- **Chain 3 (Mild Bearish Fundamental):** Layoffs and cost-cutting signal uncertainty about future growth.

3. INITIAL ASSESSMENT
The market regime is identified as **Tentative Bullish** based on technical strength. However, a significant conflict exists between the bullish macro narrative and the bearish fundamental narrative centered on AI competition. The overbought Stochastic Oscillator signals a potential for a short-term pullback.
**Agent Role:** Lead Reasoning Strategist
**Objective:** Construct a coherent narrative from the search results and classify the market regime.

> **Best Narrative Thesis:** The narrative suggests a period of macro-driven stability, where a resilient labor market and moderating inflation provide a supportive backdrop for GOOG's AI adaptation and growth initiatives.

MARKET REGIME CLASSIFICATION
- **Classification:** Weak Bullish
- **Synthesis:** The classification is derived from supportive macro data and positive sentiment around GOOG's AI advancements. Technicals show price consolidating above key support levels.

RISK/REWARD PROFILE (STL-II)

- **Structural Alignment Check: Aligned**. The implied behavior is LEANING LONG, which matches the Weak Bullish regime.

- **Final Qualitative Synthesis: Aligned but Cautious Action**.

  > **Heuristic Reference (2b53...5792):** *Cognitive Flaw: Over-reliance on short-term technical alignment, underweighting persistent fundamental and competitive bearish signals. Corrective Heuristic: IF STL-II indicates 'Aligned but Cautious' in a Weak Bullish regime, THEN before initiating a long position, conduct a weighted analysis of conflicting signals... reduce the position size by at least 50% or avoid the trade entirely, regardless of short-term technical alignment.*

## Step 3: Meta & Critique

**Agent Role:** Reviewer
**Objective:** Scrutinize the previous reasoning for logical flaws, biases, or overlooked conflicts, and propose an analytical adjustment.

> **Central Thesis for Next Cycle:** Is the current "Weak Bullish" market regime assessment for GOOG overly optimistic, given the persistent fundamental headwinds and the conflicting technical signals (strong momentum vs. overbought indicators), mirroring past instances where macroeconomic vulnerabilities were underestimated?

KEY CONFLICTING EVIDENCE

- **For (Bullish):** Strong unemployment data, price above SMAs, positive MACD.

- **Against (Bearish):** Post-earnings stock decline due to AI fears, ongoing layoffs, overbought Stochastic Oscillator.

PROPOSED ANALYTICAL ADJUSTMENT

Downgrade the confidence in the "Weak Bullish" regime. Apply a position sizing scaling factor based on a risk-reward ratio that is heavily weighted against the identified structural tensions (Macro vs. Fundamental). This is based on historical wisdom where similar regimes led to losses when macro uncertainty was underestimated.

*The Meta Agent's critique triggers a corrective reasoning loop. The system now re-evaluates its assessment with a focus on the identified conflicts.*

## Step 4: Refined Reasoning (Post-Critique)

**Agent Role:** Lead Reasoning Strategist
**Objective:** Re-assess the narrative, incorporating the Meta Agent's directive to weigh fundamental concerns more heavily.

> **Revised Narrative Thesis:** The market resilience, driven by positive sentiment and technical strength, is temporarily masking underlying fundamental concerns about GOOG's long-term growth prospects and competitive pressures.

MARKET REGIME CLASSIFICATION

- **Classification:** Weak Bullish (Maintained, but with lower confidence).

- **Synthesis:** The regime is maintained due to strong technical momentum and macro support, but the synthesis now explicitly acknowledges that this is tempered by significant fundamental headwinds.

RISK/REWARD PROFILE (STL-II)

- **Structural Alignment Check: Aligned**.

- **Final Qualitative Synthesis: Aligned but Cautious Action**.

> **Heuristic Reference (a333...0fe6):** *Prioritizing fundamental catalysts over short-term technical overbought signals. Corrective Heuristic: IF the market regime is neutral or weakly bullish AND high-impact positive fundamental catalysts are present AND technical indicators show overbought conditions, THEN prioritize the fundamental catalysts as dominant signals. Down-weight the short-term overbought signals.*

## Step 5: Final Meta & Directive

**Agent Role:** Reviewer
**Objective:** Perform a final check, focusing on the tactical implementation and signal priority.

> **Central Thesis for Next Cycle:** Is the current market's strong upward momentum a more reliable indicator of GOOG's short-term price movement than the overbought Stochastic Oscillator, and should position sizing be adjusted accordingly?

PROPOSED ANALYTICAL ADJUSTMENT (HEURISTIC)

Given the conflicting signals of strong upward momentum and an overbought Stochastic Oscillator in a Weak Bullish regime, **prioritize the overall market momentum as the dominant short-term signal**. Reduce the weighting of the Stochastic Oscillator in the final assessment. Adjust position sizing to reflect higher confidence in the upward trend, but still apply a risk-reduction factor due to the persistent fundamental concerns (AI competition, etc.). This balances seizing the opportunity with prudent risk management, as learned from historical cases.

## Step 6: Exit Workflow & Final Decision

**Agent Role:** Active Alpha Trader
**Objective:** Execute a final, quantifiable trading action based on the full reasoning process.

```
{
    "Reasoning": "The current situation is 'Perfect Alignment'.
    ↪   The Market Regime is Weak Bullish, and STL-II indicates
    ↪   a leaning long position. This alignment suggests a
    ↪   high-quality trading signal. The base confidence is set
    ↪   to Medium due to the Weak Bullish regime. STL-II's tag
    ↪   'Aligned but Cautious Action' means confidence remains
    ↪   Medium. The final decision is based on a high-quality
    ↪   aligned signal, but with 'Cautious' tag.",
    "Decision": "BUY",
    "Quantity": 6580,
    "Confidence": "Medium"
}
```

SUMMARY OF ACTION

The system concludes that despite underlying risks, the alignment of the market regime with the implied long behavior, backed by strong momentum indicators, constitutes a valid trading signal. It issues a **BUY** order with medium confidence, with the quantity likely scaled down due to the "Cautious" flag raised during the reasoning process.

### A.11 ETHICS STATEMENT AND AI USAGE DISCLOSURE

- **Ethics and Societal Impact**: The FinThink framework presented in this paper is an academic research prototype, designed to explore the capabilities of LLM-based multi-agent systems in financial reasoning within a controlled, simulated environment. It is crucial to emphasize that FinThink is not intended for real-world deployment as a financial advisory tool or automated trading system. Financial markets are subject to extreme complexity, volatility, and unpredictable events that may not be fully captured by the historical data used in our backtests. Therefore, any outputs generated by the system should be considered illustrative and should under no circumstances be interpreted as investment advice. Using this system for actual trading could lead to significant financial loss, and we strongly advise against it.

- **Use of Large Language Models in Paper Preparation**: In accordance with the ICLR 2026 policy, we disclose that Large Language Models (LLMs) were utilized to aid and polish the writing of this manuscript. The primary use was to improve grammatical correctness, clarity, and overall readability. The core scientific contributions, including the proposed framework, experimental design, data analysis, and conclusions, are entirely the original work of the authors. The authors take full responsibility for the final content of this paper.

