# OpenReview forum: "FinThink: An LLM-based Multi-agent System for Financial Reasoning"
_ICLR.cc/2026/Conference — Submitted to ICLR 2026_

### Official Review · Reviewer_PbdE · 2025-11-01

**Soundness:** 3
**Presentation:** 3
**Contribution:** 3
**Rating:** 8
**Confidence:** 4

**Summary:**

This paper introduces FinThink, a LLM-based multi-agent system designed for financial reasoning and trading. The authors posit that existing financial MAS fail because their static workflows are misaligned with the dynamic, evolving nature of real-world markets, a concept described by the Adaptive Markets Hypothesis.
To solve this, FinThink introduces three core innovations:
(1) Context-aware Workflow for Reasoning. An adaptive architecture where an LLM planner proposes reasoning steps and depth based on market signal complexity, which is then validated by a deterministic Finite State Machine (FSM). This allows the system to "think deeper" during periods of uncertainty.
(2) Reasoning-Driven Hierarchical Memory. An "evolutionary" memory system that moves beyond simple PnL triggers. It reflects on entire trade lifecycles to distill "corrective heuristics." Critically, it organizes these memories into "Narrative Driver Clusters" to enable cross-asset learning.
(3) Sentiment-To-Logic Prompt Protocol. A structured prompting strategy that decouples analysis from decision-making. Analytical agents are constrained to produce objective, logical assessments, preventing the system from collapsing into a simplistic "vote" between "BUY" or "SELL".

**Strengths:**

The paper's foundation in the Adaptive Markets Hypothesis is a significant strength. It provides a clear theoretical justification for why existing static models fail and what a successful system needs. This moves the work beyond simple engineering and into sound scientific reasoning.
The Reasoning-Driven Hierarchical Memory is the standout contribution. The idea of learning from entire trade lifecycles rather than reactive, trigger-based reflections, like a simple PnL drop, is far more robust. The concept of "Narrative Driver Clusters" to facilitate cross-asset generalization is particularly insightful and well-argued.
The ablation study in Table 3 is excellent. It clearly demonstrates the critical importance of both the R-Mem (performance is catastrophic without it) and the STL protocol (performance severely degrades without it). This provides strong quantitative support for the authors' design choices.

**Weaknesses:**

The backtest period (2022-2023) is well-chosen for its volatility but represents only one type of market (high-inflation, rate-hike-driven). The architecture's performance is not validated in other, equally common regimes, such as a sustained, low-volatility bull market (e.g., 2017) or a sudden crisis-driven crash (e.g., 2008 or Mar 2020). It is possible the system's "adaptive" nature is overfit to the specific type of volatility seen in the test window. Even though for that period the author may not be able to compare with the SOTA, you can compare with the market and simple strategy.

**Questions:**

On CWRM Hyperparameters: Figure 4 shows the system's reasoning depth, sometimes hitting a "max round hit" (which appears to be 15 rounds from the graph). How was this maximum depth chosen? Is it a fixed hyperparameter, and how sensitive is the system's performance (and, just as importantly, its computational cost/latency) to this value?

How is a "trade lifecycle" practically defined for the R-Mem reflection? Is it simply from the OPEN to CLOSE of a given position? How does the system handle more complex scenarios like partially closing a position or scaling into a position over time?

---

> ### Author Response · Authors · 2025-11-23
> **Response to Reviewer PbdE: Appreciating Feedback and Clarifications**
>
> **Thank you very much for your high recognition of our work!**
>
> Regarding the several questions you raised, I have provided the following point-by-point responses.
>
> **Re: Question 1 (Max Round Hit and Trade-offs):**
> - The Max Round Hit is a trade-off we established after observing system behavior. While computational cost and latency are factors, the primary reason is to prevent analysis paralysis. In extremely complex market scenarios, the agent system may tend to enter an infinite loop of Reasoning-Meta-Reasoning, hesitating like a human without producing marginally better reasoning outcomes. In such cases, increasing token consumption and forcing more rounds results in diminishing returns.
>
> **Re: Question 2 (Trade Lifecycle Definition):**
>
> A Trade Lifecycle is defined as a complete Open-to-Close cycle.
>
> - Scaling In: We utilize a FIFO (First-In-First-Out) mechanism. If the agent scales into a position (adds to it), R-Mem treats the subsequent closing of positions sequentially based on when they were opened.
>
> - Partial Closing: If a position is partially closed, the system locks the logs associated with that specific tranche of the trade (including the rationale used at opening, the specific CWRM outputs, and the market state at opening). The full reflection and memory generation occur only once that specific tranche is fully closed.
>
> **Re: Weakness (Market Regimes & Backtest Period):**
>
> - We appreciate you highlighting the limitation regarding the specific market regime. Indeed, in a unidirectional Bull Market (e.g., 2017), a simple Buy & Hold strategy often outperforms complex systems, and even a random agent can generate Beta returns. Such environments fail to distinguish whether an agent is intelligent or simply lucky. We selected the 2022-2023 period precisely because it represents a Hell Mode—characterized by the double killing of longs and shorts, and conflicts between macro signals and stock-specific signals. This volatility is the true litmus test for Agent intelligence (Alpha).
>
> - Additionally, regarding reproducibility: most public financial API providers limit granular historical data access to the most recent 3 years. To ensure future researchers can reproduce our work, we aligned our testing window with this accessible recent history constraint, which coincidentally covers this critical high-volatility period.
>
> Once again, we sincerely appreciate your recognition and valuable feedback! Really hope you can have a nice day ! :)

---

### Official Review · Reviewer_kuPQ · 2025-11-01

**Soundness:** 2
**Presentation:** 3
**Contribution:** 2
**Rating:** 2
**Confidence:** 4

**Summary:**

This paper introduces FinThink, an LLM-based multi-agent system designed for adaptive financial reasoning grounded in the Adaptive Markets Hypothesis (AMH). The framework integrates three key components: a Context-Aware Workflow for Reasoning (CWRM) that dynamically adjusts reasoning depth based on market uncertainty, a Reasoning-Driven Hierarchical Memory (R-Mem) that captures cross-asset learning through Narrative Driver Clusters, and a Sentiment-to-Logic (STL) prompt protocol that prevents shallow “voting-style” reasoning. Backtests on five major technology stocks are conducted based on key financial metrics. The paper argues that FinThink’s modular adaptivity and reflective memory design enable stable, explainable decision-making under dynamic market conditions.

**Strengths:**

1) The paper is clearly written and well-structured, making a technically dense topic accessible. Each core component of the proposed framework is accompanied by informative visualizations that effectively illustrate its architecture.

2) The related work (Section 2) is comprehensive and well contextualized, providing succinct but informative introductions to prior studies across both multi-agent systems and financial LLM agents.

3) A good point is the accurate and thoughtful application of classical financial theories to the system’s design rationale. The authors are able to ground FinThink in the Adaptive Markets Hypothesis (AMH), effectively using it to motivate dynamic adaptivity rather than invoking it superficially.

**Weaknesses:**

1) While FinThink presents an elaborate architecture combining a Context-aware Workflow (CWRM), a Reasoning-driven Hierarchical Memory (R-Mem), and a Sentiment-to-Logic (STL) prompt protocol, the degree of novelty appears limited compared to existing financial multi-agent framework such as FinCon[1] and HedgeAgents[2]. The conceptual framing around adaptivity and reflective reasoning is largely incremental over these works, which already explored hierarchical memory, risk control, and multi-agent reasoning frameworks for trading decision-making.

[1] Yu, Y., Yao, Z., Li, H., Deng, Z., Jiang, Y., Cao, Y., ... & Xie, Q. (2024). Fincon: A synthesized llm multi-agent system with conceptual verbal reinforcement for enhanced financial decision making. Advances in Neural Information Processing Systems, 37, 137010-137045.

[2] Li, X., Zeng, Y., Xing, X., Xu, J., & Xu, X. (2025, May). Hedgeagents: A balanced-aware multi-agent financial trading system. In Companion Proceedings of the ACM on Web Conference 2025 (pp. 296-305).

2) The experimental outcomes summarized in the Table 1 and 2 reveal that the empirical improvements are not consistently significant across all evaluated assets and metrics. As seen in Tables 1–2 (pp. 9–10), FinThink outperforms baselines in Sharpe and Calmar Ratios on average, but its Total Return (TR) often falls below FinCon for certain stocks (e.g., AAPL, GOOG, MSFT), suggesting that the method’s advantage could lie primarily in conservative risk management rather than genuine predictive superiority. This raises concerns about whether the reported gains reflect robust reasoning enhancement or simply stronger regularization.

3) Although Appendix A.1 includes an ablation study for R-Mem and STL modules, it omits a systematic examination of the modified risk module introduced in the main text (via STL-II integration and cross-asset memory), as the authors emphasize that this is a major systematic change of their framework versus prior works. The paper does not isolate how these design elements contribute to the final performance, leaving their effectiveness short for evidences. A deeper ablation or sensitivity analysis on risk-module parameters, together with statistical significance testing, would be necessary to substantiate the claimed benefits and demonstrate the distinct contribution of each innovation.

**Questions:**

Refer to what I mentioned in the weaknesses.

---

> ### Author Response · Authors · 2025-11-23
> **Rebuttal to Reviewer kuPQ by Authors (Part 1)**
>
> Thank you very much for your insightful review. Below are my detailed responses to the points you mentioned regarding my weaknesses:
>
> **1.Re: Weakness 1 (The Novelty of  CWRM) :**
> We argue that a primary challenge in financial reasoning is the "Reasoning Load Problem"—the difficulty of identifying and addressing key reasoning challenges within a complex financial environment. To address this, we formalized the process by decomposing it into distinct modules: the Reasoning Workflow, Meta Workflow, Search Workflow, and Router Agent. This decomposition allows each workflow to handle specific sub-steps independently, effectively reducing the cognitive load of the overall task. Specifically, the Reasoning Workflow focuses on inferring complex problems, while the Meta Workflow detects and corrects flaws in that reasoning. The Router Agent then evaluates whether the current conclusion is sufficiently robust and whether the issues raised by the Meta Workflow have been resolved, ultimately guiding the process to the Exit Workflow for decision-making.
>
> As shown in our ablation study (Appendix), the lack of CWRM scheduling leads to catastrophic performance degradation. This failure stems directly from a lack of "elastic reasoning"; without systematic reflection and dynamic scheduling, agent systems tend to execute trades based on shallow analysis, leading to significant drawdowns. In short, a key contribution of FinThink is decoupling financial analysis into three distinct cognitive processes via CWRM:
>
> - **Depth Perception of Analysis**
> - **Quality Maximization**
> - **Analysis Checking**
>
> This adaptive reasoning design, grounded in the AMH, represents an approach not attempted by prior works. Furthermore, the "max round hit" observed in CWRM demonstrates the agent's ability to perceive context complexity (i.e., the "load") and allocate computational resources (inference rounds) accordingly.
>
> **2.Re: Weakness 2 (Total Return vs. Risk-Adjusted Return) :**
> Total Return (TR) is not always the superior metric for evaluation. In financial trading and probability games, the "Hot Hand Fallacy" describes scenarios where humans or AI achieve high returns over a short period purely due to luck or aggressive beta exposure. We acknowledge that FinCon [1] achieves higher raw profitability. However, we emphasize that the Sharpe Ratio is widely regarded as a more reliable metric for sustainable trading because it accounts for both return and volatility (risk). A system that ignores conservative risk management may achieve high returns temporarily but struggles to maintain stable positive performance or avoid ruin in the long run.
>
> In FinThink, we formalize trading judgments via the STL Protocol. As detailed in the Appendix, the Reasoning Workflow formalizes the Search output into three distinct criteria:
>
> - **Regime (a judgment of the aggregate market state)**
>
> - **Narrative (short-term tradeable stories)**
>
> - **STL-II (tactical rationality judgment)**
>
> The STL-II module performs a validity check based on the alignment of the Regime and Narrative. This comprehensive analysis is then fed to the decision agent. Our system's excess returns and stability stem primarily from this conscious decoupling of long-term vs. short-term signals (Regime vs. Narrative) and the STL-II gating mechanism. This capability relies on the holistic CWRM architecture, which cycles through information retrieval, reasoning, and meta-cognitive reflection.
>
>
> [1] Yu, Y., Yao, Z., Li, H., Deng, Z., Jiang, Y., Cao, Y., ... & Xie, Q. (2024). Fincon: A synthesized llm multi-agent system with conceptual verbal reinforcement for enhanced financial decision making. Advances in Neural Information Processing Systems, 37, 137010-137045.

---

> ### Author Response · Authors · 2025-11-23
> **Rebuttal to Reviewer kuPQ by Authors (Part 2)**
>
> Dear Reviewer, the previous response was the first part of our reply to the issues you raised, mainly addressing your first two weaknesses. Below is the second part, mainly responding to your third weakness:
>
> **3.Re: Weakness 3 Part One (Clarification on STL-II Effectiveness) ：**
> We must clarify that STL-II is not a tunable hyperparameter module, but rather a critical gatekeeper within the STL Prompt Protocol (see Appendix A.9). Within this protocol, we redefine risk as "Logical Divergence." STL-II permits a trade only when the medium-term Market Regime and the short-term Trading Narrative reach a consensus. Any divergence (e.g., a Bearish Regime but Bullish Narrative) is treated as a structural risk, triggering a mandatory "No Trade" or "Cover" action. Regarding the validation of its effectiveness: The ablation study in Table 3 (w/o STL) removes the entire protocol, including this gatekeeper mechanism. This resulted in a drastic drop in the Sharpe Ratio from 2.14 to 1.61. This significant statistical degradation directly confirms that this "decoupled reasoning" mechanism is the core driver of our high risk-adjusted returns.
>
> **4.Re: Weakness 3 Part Two (Questioning R-Mem & Cross-Asset Learning) :**
> We argue that the advantage of Cross-Asset Memory over Single-Asset Memory is not merely data mixing, but a paradigm shift from "Ticker-Centric" to "Logic-Centric" memory. First, Figure 6 provides strong visual empirical evidence.
> The visualization shows that assets with different attributes spontaneously emerge into distinct geometric structures:
>
> - **The TSLA/AMZN (High-Beta) group forms a compact, "Action-oriented" cluster.**
>
> - **The AAPL/MSFT (Blue-Chip) group forms a dispersed, "Fundamental-oriented" structure.**
>
> This strongly suggests that R-Mem captures and learns the distinct "Behavioral Manifolds" of different asset classes. By aggregating across assets, we construct a continuous and robust "Logical Surface," making reasoning more structured.
>
> Essentially, this is a form of logic-based data augmentation. In the limited history of a single asset, specific extreme market states (e.g., "Rate Hike Panic" or "Liquidity Dry-up") are extremely scarce samples. Relying solely on TSLA's history risks overfitting. However, since AMZN and TSLA share the "High-Beta" attribute, they exhibit high Logical Isomorphism under macro shocks. Introducing AMZN's memory effectively multiplies the "experience samples" available to TSLA for these scarce conditions. We acknowledge this is a pioneering conceptual attempt, but it aims to enable agents to learn universal market laws rather than overfitting single price paths.
>
> Thank you for taking the time to read our response and rebuttal. Wish you a happy day! :)

---

### Official Review · Reviewer_XFbk · 2025-11-01

**Soundness:** 2
**Presentation:** 2
**Contribution:** 1
**Rating:** 4
**Confidence:** 3

**Summary:**

This paper proposes FinThink, a Multi-Agent System based on the AMH theory. While its motivation (addressing static workflows) is reasonable, the proposed methods, CWRM and R-Mem, appear highly heuristic and over-engineered, lacking clear formal definitions and data-driven validation. Furthermore, the paper's primary performance claims rely heavily on risk-adjusted metrics, while the cumulative return plots in the appendix (Fig 8) show FinThink often underperforms multiple baselines in terms of Total Return. This, combined with several clarity issues and inconsistencies in key metric definitions (Sec A.4), makes the paper's contribution difficult to verify and its effectiveness questionable.

**Strengths:**

1. The paper grounds its framework in the Adaptive Markets Hypothesis (AMH), which is a strong conceptual starting point. This provides a sound justification for why the system needs "architectural adaptivity" and "evolutionary adaptivity".


2. Despite issues with total return, the system is highly successful in achieving its risk management objective.

**Weaknesses:**

1. The paper claims "significant improvements" and "superior risk-adjusted returns" in the abstract and experiments section. These claims rest entirely on risk-adjusted metrics (SR, CR) and MDD. However, the cumulative return curves in Appendix A.2 reveal a very different story: FinThink is mediocre in terms of total return. FinThink's total return is visibly lower than the B&H benchmark and other models like FinCon.

2. The core of CWRM is the "FSM Controller," which acts as a "rule enforcer" for "logical coherence". Yet, the specific state transition rules of this FSM are never defined in the paper. Algorithm 1 is just a black-box validateTransition call.  The core innovation of R-Mem, the "Narrative Driver Clusters", is admitted to be a "fundamentally a heuristic design".

3. The experiment relies on Gemini-2.0-flash only. It is important to see how FinThink works for different LLMs, at least a stronger one.

4. The experimental setup for the cross-asset memory is questionable. The paper mixes TSLA and AMZN data for training and then evaluates the model on these same assets. This methodology is poorly explained and raises serious concerns about potential data leakage, as it fails to test the model's ability to generalize to truly unseen assets.

5. The memory methodology is vague and insufficiently evaluated. The paper states the memory is trained for only 3 epochs, and the process of how "corrective heuristics" are precisely extracted from trade cycles is not clearly articulated; it reads more like a narrative than a defined algorithm. Furthermore, the memory component is not directly compared to baseline mechanisms (like FinCon's) and lacks dedicated ablation studies or case studies to validate its specific contribution.

**Questions:**

Please see the weaknesses above.

---

> ### Author Response · Authors · 2025-11-23
> **Rebuttal to Reviewer XFbk by Authors (Part 1)**
>
> Dear Reviewer, thank you very much for pointing out these weaknesses. They are very useful to us. The following is my detailed response to your questions and the weaknesses you raised:
>
> **1.Re: Weakness 1 (Total Return vs. Risk-Adjusted Metrics)**:
>
> - Total Return is not always the definitive metric for evaluation. In financial trading and probability-based games, the "Hot Hand Fallacy" describes a phenomenon where a human or AI achieves high returns over a specific period purely due to luck rather than skill. We candidly acknowledge that FinCon [1] demonstrates stronger raw profitability (maximum return capability) than our model. However, we must emphasize that the Sharpe Ratio is widely regarded as a more reliable metric in financial trading because it synthetically accounts for both returns and volatility (risk). If a system merely adopts a conservative risk management strategy without effective alpha generation, it is often difficult to achieve stable positive returns—in fact, such strategies often lead to stagnation or losses.
>
> - In our system, we formalize trading judgments via the STL Protocol. As detailed in the Appendix, the Reasoning Workflow transforms the output of the Search Workflow into three core criteria: Market Regime, Core Narrative, and STL-II (Tactical Rationality Check). Here, the Narrative represents ultra-short-term tradeable stories, while the Regime represents a holistic,  judgment of all aggregated market information. STL-II then performs a tactical validity check based on the alignment of the Regime and Narrative. Upon combining these three elements, a highly comprehensive analytical report is fed to the final Decision Agent for execution.
>
>  - Our system's excess returns and stability stem primarily from this conscious decoupling of long-term and short-term signals (Regime-Narrative) and the gating function of STL-II. Essentially, realizing this capability depends on the holistic CWRM architecture, which decouples the inference process into three independent modules—Information Search/Integration, Reasoning, and Meta-Cognitive Reflection—and coordinates them via iterative loops. The improvement in Sharpe Ratio and the control of Maximum Drawdown are direct results of the synergistic cooperation between CWRM and the STL Prompt Protocol.
>
> **2. Re: Weakness 2 (Formalization of FSM):** To address your concern regarding the formal definition of the FSM scheduling, we provide the algorithmic description below:
>
> **Algorithm: Formal Definition of the Finite State Machine (FSM) for CWRM**
>
> **Require:** Current State $S_t$, Workflow Proposal $S_{prop}$, Context $C_t$, Max Usage Constraints $U_{max}$
>
> **Ensure:** Valid Next State $S_{t+1}$, Transition Signal $\delta \in \{Valid, Invalid\}$
>
> 1. **Define State Space:** $S = \{Start, Search, Reasoning, Meta, Exit\}$
> 2. **Define Transition Function:** $T: S \times S \rightarrow \{0, 1\}$
> 3. **Initialize:** $S_0 \leftarrow Start$, Counters $Count[s] \leftarrow 0$ for all $s \in S$
>
> 4. **Function** ValidateTransition($S_t, S_{prop}, Count, Round$)
>    - $\delta \leftarrow Invalid$
>
>    - **If** $S_t ==  start $ **AND** $S_{prop} \neq  search $ **Then**
>      - **Return** $(S_t, Invalid)$  // Only transition from start to search is allowed, because without information collection, any reasoning is meaningless.
>    - **End If**
>
>    // Enforce maximum workflow usage (4 times per workflow)
>    - **If** $Count[S_{prop}] \ge 4$ **Then**
>      - **Return** $(S_t, Invalid)$  // Workflow has reached its maximum usage
>    - **End If**
>
>    // Enforce maximum rounds (15 rounds)
>    - **If** $Round \ge 15$ **Then**
>      - **Return** $(S_t, Invalid)$  // Maximum round limit reached
>    - **End If**
>
>    // Enforce resource constraints
>    - **If** $Count[S_{prop}] \ge U_{max}[S_{prop}]$ **Then**
>      - **Return** $(S_t, Invalid)$  // Enforce resource constraints
>    - **End If**
>
>    // If everything is valid
>    - $S_{t+1} \leftarrow S_{prop}$
>    - $Count[S_{t+1}] \leftarrow Count[S_{t+1}] + 1$
>    - **Return** $(S_{t+1}, Valid)$
>
> 5. **End Function**
>
> 6. **Main Execution Loop:**
>    - **While** $S_t \neq Exit$ **AND** $Round < N_{max}$ **Do**
>      - $S_{prop} \leftarrow \text{RoutingAgent}(Context_t)$
>      - $(S_{next}, \delta) \leftarrow \text{ValidateTransition}(S_t, S_{prop}, Count, Round)$
>      - **If** $\delta == Valid$ **Then**
>        - $S_t \leftarrow S_{next}$
>        - Execute Workflow corresponding to $S_t$
>        - Update $Context_{t+1}$ with workflow output
>      - **Else**
>        - Trigger Re-planning or Error Handling
>      - **End If**
>    - **End While**

---

> > ### Author Response · Authors · 2025-11-23
> > **Rebuttal to Reviewer XFbk by Authors (Part 2)**
> >
> > **3. Re: Weakness 2 (Explanation of R-Mem):** Regarding R-Mem, we respectfully clarify that it is not a highly heuristic design, but rather a deliberate architectural choice derived from deep analysis. We posit that the advantage of Cross-Asset Memory over Single-Asset Memory lies not in simple data mixing, but in a paradigm shift from "Ticker-Centric" to "Logic-Centric" learning.
> >
> > - **Figure 6** visually demonstrates that asset groups with distinct characteristics form contrasting geometric structures. The TSLA/AMZN (High-Beta) group shows a compact, "Action-oriented" structure, while the AAPL/MSFT (Blue-Chip) group displays a dispersed, "Fundamental-oriented" structure. This confirms that R-Mem captures the unique "Behavioral Manifolds" of these asset classes and constructs a continuous, structured "Logical Surface" for reasoning.
> >
> > - This mechanism also functions as data augmentation. Extreme market regimes (e.g., "Rate Hike Panic") are rare for individual assets like TSLA, making it hard for the agent to generalize. However, since AMZN and TSLA share the High-Beta attribute, their logical isomorphism allows memory sharing, effectively increasing the agent's sample size for these rare market conditions. This approach aims to help the agent learn universal market dynamics, avoiding overfitting to a single asset's historical price path.
> >
> > **4. Re: Weakness 3 (Gemini-2.0-flash Limitations):** We appreciate you raising the limitation regarding gemini-2.0-flash. Indeed, testing a single model class may seem insufficient. However, we deliberately chose gemini-2.0-flash because it is a lightweight model optimized for cost rather than reasoning power. In recent public AI trading competitions, the Gemini series has often ranked lower in raw trading performance compared to flagship models like GPT-4. By achieving state-of-the-art results using a "weaker," cost-effective model, we demonstrate that the performance gains stem from our architecture (FinThink) rather than the raw capabilities of the underlying LLM.
> >
> > **5. Re: Weakness 4 (Data Leakage Concerns):** We employ a strict Temporal Split for the training and testing phases. The memory training period is entirely separate from the backtesting period. Formal backtesting only begins after memory training concludes, ensuring strict information isolation and preventing any look-ahead bias.
> >
> > **6. Re: Weakness 5 (Comparison with FinCon):** We acknowledge FinCon[1] as a pioneering work in LLM-based financial agents. However, we respectfully clarify that a direct module-level comparison (e.g., swapping memories) is challenging due to fundamental architectural distinctions rather than mere performance differences.
> > - **Design Philosophies Comparison:** First, regarding design philosophies, FinCon[1] excels in offline policy optimization. It effectively utilizes statistical indicators (CVaR) and outcome-based feedback (PnL) to refine agent prompts during the training phase, treating risk management primarily as a statistical gating task. In contrast, FinThink focuses on runtime architectural adaptivity via CWRM. Recognizing that financial markets under the AMH are dynamic and noisy, we prioritize "Proactive Ambiguity Detection" over reactive outcome feedback. Our system dynamically adjusts reasoning depth before a trade is placed, aiming to filter out stochastic noise that might be misidentified as valid signals in a purely outcome-driven loop.
> > - **Structural Divergence in Memory:** Second, regarding structural divergence in memory, while FinCon[1] employs a selective propagation mechanism for belief updates, FinThink introduces "Narrative Driver Clusters" to structurally operationalize cross-asset generalization. This represents a topological shift—moving from optimizing individual agent prompts (as in FinCon[1]) to constructing a shared, semantic knowledge base that transfers heuristics between logically connected assets.
> >
> > Therefore, FinThink should be viewed as an evolution that extends financial agents from "Statistical Risk Control" (FinCon[1]) to "Logical Risk Reasoning" (FinThink), addressing the specific challenges of reasoning load and noise robustness in a distinct manner.
> >
> > We appreciate your feedback and the time you took to review our work. Hope you can have a good day! :)
> >
> > [1] Yu, Y., Yao, Z., Li, H., Deng, Z., Jiang, Y., Cao, Y., ... & Xie, Q. (2024). Fincon: A synthesized llm multi-agent system with conceptual verbal reinforcement for enhanced financial decision making. Advances in Neural Information Processing Systems, 37, 137010-137045.

---

### Meta-Review · Area_Chair_mxAN · 2026-01-07

**Summary:**

This paper proposes FinThink, a Multi-Agent System based on the adaptive markets hypothesis theory. Reviewer's major concerns include (1) multiple reviewers questioned the choice of performance evaluation metrics. (2) the state transition rules of the FSM controller is not defined in the paper (3) experiments are based on Gemini 2.0 flash only (4) lack of clarity in memory methodology (5) limited novelty comparing to existing financial multi-agent framework (6) lack of evaluation of the modified risk module. (7) questions on generalizability to different types of market.

**Reviewer Concerns:**

Concerns (1) (2) (4) (7) are addressed. the author did not provide additional experiments to address (3) (6). The novelty arguments made by the authors for (5) is not convincing.

**Reviewer Scores:**

Reviewer XFbk might increase his score. The other two reviewers might keep their original scores.

---

### Decision · Program_Chairs · 2026-01-26

Reject